# META-LEARNING ADAPTABLE FOUNDATION MODELS

## ABSTRACT

The power of *foundation models* (FMs) lies in their capacity to learn highly expressive representations that can be adapted to a broad spectrum of tasks. However, these pretrained models require multiple stages of fine-tuning to become effective for downstream applications. Conventionally, the model is first retrained on the aggregate of a diverse set of tasks of interest and then adapted to specific low-resource downstream tasks by utilizing a parameter-efficient fine-tuning (PEFT) scheme. While this procedure seems reasonable, the independence of the retraining and fine-tuning stages causes a major issue, as there is no guarantee the retrained model will achieve good performance post-fine-tuning. To explicitly address this issue, we introduce a meta-learning framework infused with PEFT in this intermediate retraining stage to learn a model that can be easily adapted to unseen tasks. For our theoretical results, we focus on linear models using low-rank adaptations. In this setting, we demonstrate the suboptimality of standard retraining for finding an adaptable set of parameters. Further, we prove that our method recovers the optimally adaptable parameters. We then apply these theoretical insights to retraining the RoBERTa model to predict the continuation of conversations between different personas within the ConvAI2 dataset. Empirically, we observe significant performance benefits using our proposed meta-learning scheme during retraining relative to the conventional approach.

## 1 INTRODUCTION

*Foundation Models* (FMs) learn rich representations that are useful for a variety of downstream tasks. FMs are trained in three general stages to fit user-specific tasks like context-specific language generation and personalized image synthesis, among others. The first stage is commonly referred to as pretraining, where FMs are trained from scratch on a combination of massive public, propriety, and synthetic sources of data to learn a general-purpose model (Devlin et al., 2019; Brown et al., 2020; Abdin et al., 2024; Radford et al., 2021). This stage is largely inaccessible to most due to the enormous cost of training state-of-the-art models on such large datasets.

Thus, the most popular and viable way to utilize FMs for individual tasks is to take a pretrained model and *retrain* it for a specific objective. In this second training stage, we refine the pretrained model and retrain it on a large set of tasks of interest. For clarity, we generally refer to this intermediate stage as retraining. Other works have referred to this stage as pre-finetuning (Aghajanyan et al., 2021) or supervised fine-tuning (Dong et al., 2024). In the third stage, referred to as fine-tuning, the model is ultimately trained on an individual low-resource task. For example, a pretrained large language model (LLM) can be downloaded and retrained on a large multi-lingual corpus to perform English-Spanish and English-Italian translations. Then, one may adapt the model to translate English to French using a small English-French translation dataset. For this last stage, the model is typically fine-tuned using parameter efficient fine-tuning (PEFT) methods – training heuristics which sacrifice learning expressiveness for improved computational efficiency (Hu et al., 2021; Li & Liang, 2021). PEFT is especially useful in the low-resource setting, as running full fine-tuning of the model's parameters on a small number of samples is expensive and potentially unnecessary.

Conventional retraining updates either a subset or all of the model parameters to fit the aggregation of the different retraining tasks. While this approach seems reasonable and has been successful in improving downstream task performance (Khashabi et al., 2020; Raffel et al., 2020), it does not leverage knowledge of the downstream fine-tuning procedure to cater the retrained model to perform well after such adaptation. Rather, it retrains the model to minimize the average loss across

the retraining tasks regardless of the PEFT method to be employed later. This raises two key issues. Firstly, there may not exist a single set of model parameters that simultaneously fits the various retraining tasks. Secondly, even if the model is sufficiently over-parameterized, there is no assurance the recovered retrained solution is indeed adaptable to future unseen tasks relative to other possible solutions during retraining, as the retraining and fine-tuning are performed independently.

We address these issues by drawing upon ideas from *meta-learning*, a framework designed to explicitly train models for future adaptation. Meta-learning is a common method to improve model performance after fine-tuning, typically in low-resource, few-shot settings using gradient-based adaptations (Finn et al., 2017; Lee & Choi, 2018). Moreover, it has begun to be applied to FM retraining to prepare models for downstream fine-tuning (Hou et al., 2022; Hong & Jang, 2022; Bansal et al., 2022; Gheini et al., 2022; Hu et al., 2023). However, it is not yet understood whether meta-learning how to fine-tune can provably confer performance benefits over standard retraining followed by PEFT. In this work, we provide rigorous theoretical and empirical evidence that this is indeed the case. We first study a stylized linear model where the ground truth parameters for both the retraining and fine-tuning tasks are realizable by low-rank adaptations. We validate our theory through synthetic data and show that our insights improve performance on real language tasks using large language models (LLMs). Specifically, our contributions are as follows:

- We develop a generalized framework to model standard retraining and propose the Meta-Adapters objective for retraining, a meta-learning-inspired objective function for infusing PEFT in foundation model retraining. Our framework can be implemented with any PEFT algorithm, but we emphasize the incorporation of LoRA (Hu et al., 2021).

- For a linear model applied to multiple tasks whose ground truth parameters are realizable by LoRA, we show standard retraining does not recover an adaptable set of model parameters (Theorem 1) and thus incurs significant loss on unseen tasks after fine-tuning (Corollary 1). We prove two key results for the Meta-Adapter's objective function:

  - Any model that globally minimizes this objective can be exactly fine-tuned to unseen tasks (Theorem 2), and when retraining on three or more tasks, the ground truth parameters are the unique global minimum up to orthogonal symmetry (Theorem 3). This uniqueness property holds as long as the data dimension is sufficiently large, which is counterintuitive to previous work on multi-task learning theory that requires the number of tasks to be larger than the effective task dimension (Du et al., 2021; Collins et al., 2022).
  - For two retraining tasks, second-order stationarity is sufficient to guarantee global minimization for our Meta-Adapters loss (Theorem 4). In this case, our Meta-Adapters objective function is provably amenable to local optimization methods.

- To test our theoretical insights, we compare the performance of the standard retraining and Meta-Adapters objectives for linear models using LoRA while relaxing the assumptions from our theory. We show clear improvements using the Meta-Adapters objective for all data generation parameter settings and for different numbers of tasks. Then, we apply our meta-learning method to the RoBERTa (Liu et al., 2019) large language model (LLM) on the ConvAI2 dataset (Dinan et al., 2019), a real-world multi-task dataset for generating continuations of conversations between different personas. Again, we show improvements using the Meta-Adapters relative to retraining then fine-tuning.

## 1.1 RELATED WORK

Meta-learning is a framework for learning models that can be rapidly adapted to new unseen tasks by leveraging access to prior tasks during training. For example, Model-Agnostic Meta-Learning (MAML) (Finn et al., 2017) is a popular, flexible method that aims to find a model that can be adapted to a new unseen task after a small number of steps of gradient descent on the unseen task's loss function. Further, other works have proposed methods specific to low-dimensional linear models and have shown strong results and connections between meta-learning and representation learning (Collins et al., 2022; Thekumparampil et al., 2021).

In the case of FMs, other lines of work have proposed meta-learning approaches where the task-specific adaptation incorporates PEFT methods rather than few-shot gradient updates of all model parameters. (Hong & Jang, 2022; Bansal et al., 2022; Gheini et al., 2022) apply meta-learning with

architecture adaptations that inject small task-specific trainable layers within the FM architecture. (Hou et al., 2022) further combines architecture adaptations with parameter perturbation adaptations similar to LoRA. They consider a complicated meta-learning loss that separates the available training tasks data into training and testing tasks, and they update the adapters and FM weights over different splits of the data. Using combinations of architecture and parameter adaptation methods, they show empirical gains over retraining, then fine-tuning, and other gradient-based MAML-style algorithms. (Aghajanyan et al., 2021) similarly proposes a multi-task objective that trains an FM on different tasks simultaneously to encourage learning a universally applicable representation. They force the FM to learn a common shared data representation and apply a different prediction head for each retraining task. They run extensive empirical studies and observe performance improvements in a large-scale setting when 15 or more tasks are used in the retraining stage.

These works propose some kind of meta-learning or multi-task objective and show empirical gains over standard retraining strategies on natural language datasets, yet none explain when standard retraining is insufficient relative to meta-learning and multi-task approaches, how many tasks are needed to learn a rich representation, and how to best adapt to tasks unseen in the training stage.

Lastly, although we focus on LoRA, different PEFT methods have been proposed, including variants of LoRA (Liu et al., 2024; Dettmers et al., 2023; Zhang et al., 2023) and architecture adaptations (Houlsby et al., 2019) among others. Further, recent works have begun analyzing theoretical aspects of LoRA in the fine-tuning stage (Jang et al., 2024; Zeng & Lee, 2023). These works have started advancing the theory of LoRA, but they explore orthogonal directions to the analysis of meta-learning infused with LoRA. We include an extended discussion of these works in Appendix A.

**Notation.** We use bold capital letters for matrices and bold lowercase letters for vectors. $\mathcal{N}(\boldsymbol{\mu}, \boldsymbol{\Sigma})$ refers to the multivariate Gaussian distribution with mean $\boldsymbol{\mu}$ and covariance matrix $\boldsymbol{\Sigma}$. $\|\cdot\|_F$ refers to the Frobenius norm. $S_d$ refers to the set of $d \times d$ symmetric matrices, and $S_d^+ = \{\boldsymbol{X} \in S_d | \boldsymbol{X} \succcurlyeq \boldsymbol{0}\}$ is the set of $d \times d$ symmetric positive semi-definite matrices. $O_d$ refers to the set of $d \times d$ orthogonal matrices. $[n]$ refers to the set $\{1, \ldots, n\}$. For a matrix $\boldsymbol{X}$, $\text{im}(\boldsymbol{X})$ and $\ker(\boldsymbol{X})$ refer to the image and kernel of $\boldsymbol{X}$, respectively. For subspaces $\boldsymbol{M}, \boldsymbol{N}$, $\dim(\boldsymbol{M})$ refers to the dimension of $\boldsymbol{M}$ and $\boldsymbol{M} + \boldsymbol{N} = \{\boldsymbol{x} + \boldsymbol{y} | \boldsymbol{x} \in \boldsymbol{M}, \boldsymbol{y} \in \boldsymbol{N}\}$. If $\boldsymbol{M} \cap \boldsymbol{N} = \{\boldsymbol{0}\}$, then we write the direct sum $\boldsymbol{M} \oplus \boldsymbol{N}$.

## 2 FOUNDATION MODEL RETRAINING AND FINE-TUNING

In this section, we first briefly recap the optimization process for conventional retraining of a foundation model (FM) across multiple tasks, followed by its fine-tuning on a downstream task. We then introduce our meta-learning-based approach which adjusts the retraining phase to incorporate insights from the final fine-tuning procedure.

### 2.1 STANDARD RETRAINING THEN FINE-TUNING

Consider a collection of $T$ tasks of interest $\mathcal{T} = \{\mathcal{T}_t\}_{t=1}^T$ where each task $\mathcal{T}_t$ is drawn from task distribution $\mathcal{D}$ and consists of $n_t$ labeled examples $\mathcal{T}_t = \{(\boldsymbol{x}_{t,j}, \boldsymbol{y}_{t,j})\}_{j=1}^{n_t}$, where $(\boldsymbol{x}_{t,j}, \boldsymbol{y}_{t,j})$ are i.i.d. from the $t_{th}$ task's data distribution $\mathcal{D}_{\mathcal{T}_t}$. Without loss of generality we assume that for all tasks $\mathcal{T}_t$ drawn from $\mathcal{D}$, $\mathcal{D}_{\mathcal{T}_t}$ generates samples $\boldsymbol{x}_{t,j} \in \mathbb{R}^{d_x}$, $\boldsymbol{y}_{t,j} \in \mathbb{R}^{d_y}$ for all $t \in [T], j \in [n_t]$. Consider a model $\Phi(\cdot\,; \boldsymbol{W}) : \mathbb{R}^{d_x} \to \mathbb{R}^{d_y}$ parameterized by weights $\boldsymbol{W}$ that maps feature vectors $\boldsymbol{x} \in \mathbb{R}^{d_x}$ to predicted labels $\hat{\boldsymbol{y}} \in \mathbb{R}^{d_y}$. Typically $\boldsymbol{W} = (\boldsymbol{W}_1, \ldots, \boldsymbol{W}_m)$ is a list of matrices where $\boldsymbol{W}_i \in \mathbb{R}^{d \times d}$ parameterize the layers of a neural network. We assume each $\boldsymbol{W}_i$ is square for convenience.

**Retraining Phase.** Given a loss function $\mathcal{L}$, standard retraining attempts to minimize the aggregated loss over a collection of training tasks (Liu et al., 2019; Brown et al., 2020). This amounts to solving

$$\min_{\boldsymbol{W}} \sum_{t=1}^T \sum_{j=1}^{n_t} \mathcal{L}\left(\Phi(\boldsymbol{x}_{t,j}; \boldsymbol{W}), \boldsymbol{y}_{t,j}\right). \tag{1}$$

In other words, the above optimization problem seeks a set of universal parameters that define a unique mapping function capable of translating inputs to outputs across all tasks involved in the retraining phase. We denote the set of weights obtained by solving (1) as $\hat{\boldsymbol{W}}_{\text{SR}}$, and the corresponding input-output mapping function as $\Phi(\cdot\,; \hat{\boldsymbol{W}}_{\text{SR}})$, where SR stands for Standard Retraining.

**Fine-Tuning Phase.** In the subsequent fine-tuning step, we refine either the retrained weights, the model's feature map, or both to fit a downstream task with fewer labeled samples. More precisely, consider a downstream task $\mathcal{T}_{T+1}$ drawn from the same distribution $\mathcal{D}$ where $\mathcal{T}_{T+1} = \{(\boldsymbol{x}_{T+1,j}, \boldsymbol{y}_{T+1,j})\}_{j=1}^{n_{T+1}}$. To fit the model to task $\mathcal{T}_{T+1}$ we do not retrain the retrained weights $\hat{\boldsymbol{W}}_{\text{SR}}$, but instead fine-tune the mapping $\Phi(\,\cdot\,; \hat{\boldsymbol{W}}_{\text{SR}})$ using additional parameters $\boldsymbol{\theta}$. For example, $\boldsymbol{\theta}$ could parameterize transformations of $\hat{\boldsymbol{W}}_{\text{SR}}$ that adapt the retrained weights or new trainable layers inserted into the architecture of the retrained model (Hu et al., 2021; Liu et al., 2024; Aghajanyan et al., 2021). We denote the fine-tuned model's mapping as $\Phi_{\text{FT}}(\,\cdot\,; \hat{\boldsymbol{W}}_{\text{SR}}, \boldsymbol{\theta}) : \mathbb{R}^{d_x} \to \mathbb{R}^{d_y}$. During the *fine-tuning stage*, the goal is to find the optimal additional parameters, $\boldsymbol{\theta}$, that minimize the loss for the downstream task $\mathcal{T}_{T+1}$, solving:

$$\min_{\boldsymbol{\theta}} \sum_{j=1}^{n_{T+1}} \mathcal{L}(\Phi_{\text{FT}}(\boldsymbol{x}_{T+1,j}\,; \hat{\boldsymbol{W}}_{\text{SR}}, \boldsymbol{\theta}), \boldsymbol{y}_{T+1,j}). \tag{2}$$

In particular, when the LoRA PEFT method is used for fine-tuning, the model is adapted to task $\mathcal{T}_{T+1}$ by fixing the model architecture and the retrained weights $\hat{\boldsymbol{W}}_{\text{SR}}$ and only training low-rank perturbations for each of the matrices $\hat{\boldsymbol{W}}_{\text{SR},1}, \ldots, \hat{\boldsymbol{W}}_{\text{SR},m}$. For rank-$k$ adaptations, we parameterize $\boldsymbol{\theta} = ((\boldsymbol{U}_1, \boldsymbol{V}_1), \ldots, (\boldsymbol{U}_m, \boldsymbol{V}_m))$, where $\boldsymbol{U}_i, \boldsymbol{V}_i \in \mathbb{R}^{d \times k}$ are the factors of the low-rank adaptation of the $i$th matrix in $\hat{\boldsymbol{W}}_{\text{SR}}$. The fine-tuned model is just the original model where the $i$th weight matrix $\boldsymbol{W}_i$ is now perturbed to be $\boldsymbol{W}_i + \boldsymbol{U}_i \boldsymbol{V}_i^{\top}$. Then the LoRA fine-tuning optimization problem is:

$$\min_{\{\boldsymbol{U}_i, \boldsymbol{V}_i\}_{i=1}^m} \sum_{j=1}^{n_{T+1}} \mathcal{L}\left(\Phi\left(\boldsymbol{x}_{T+1,j}\,; \left(\hat{\boldsymbol{W}}_{\text{SR},1} + \boldsymbol{U}_1 \boldsymbol{V}_1^{\top}, \ldots, \hat{\boldsymbol{W}}_{\text{SR},m} + \boldsymbol{U}_m \boldsymbol{V}_m^{\top}\right)\right), \boldsymbol{y}_{T+1,j}\right). \tag{3}$$

This pipeline seems reasonable as we first fit the model to the aggregation of the retraining tasks which we hope will promote learning the general structure of the tasks drawn from $\mathcal{D}$. However, there may not exist a single model that can model each retraining task simultaneously, so retraining the model on the aggregation of the retraining tasks does not align with our implicit assumption that each task is realizable after task-specific adaptations from a common model. Further, even if the model is sufficiently overparameterized where many possible solutions fit the retraining tasks, standard retraining finds a solution independent of the subsequent PEFT method to be used for fine-tuning. Nothing about standard retraining promotes learning an adaptable solution relative to other candidate solutions that fit the retraining tasks.

## 2.2 META-ADAPTERS

Since the ultimate goal of our model is to perform well on a variety of unseen downstream tasks, we propose the Meta-Adapters objective that explicitly fits weights and adapter parameters to the training tasks. Intuitively, this objective promotes sets of parameters that can be adapted to future unseen tasks drawn from the same distribution as those seen in retraining.

Rather than training a single model on the aggregation of the retraining tasks, we instead incorporate the adapters during the retraining process and learn adapted models for each task. Let $\boldsymbol{\theta}^{(t)}$ be the set of adapter parameters for the $t_{th}$ training task $\mathcal{T}_t$. The Meta-Adapters method searches for a single set of base weights $\hat{\boldsymbol{W}}_{\text{Meta}}$ such that for all $t \in [T]$, the $t_{th}$ adapted model $\Phi_{\text{FT}}(\,\cdot\,; \hat{\boldsymbol{W}}_{\text{Meta}}, \boldsymbol{\theta}^{(t)})$ minimizes the loss over the training task $\mathcal{T}_t$. We define the Meta-Adapters objective as:

$$\min_{\boldsymbol{W}} \sum_{t=1}^{T} \min_{\boldsymbol{\theta}^{(t)}} \sum_{j=1}^{n_t} \mathcal{L}\left(\Phi_{\text{FT}}\left(\boldsymbol{x}_{t,j}\,; \boldsymbol{W}, \boldsymbol{\theta}^{(t)}\right), \boldsymbol{y}_{t,j}\right). \tag{4}$$

When we use LoRA as the adaptation method, we define $\boldsymbol{U}_i^{(t)}\left(\boldsymbol{V}_i^{(t)}\right)^{\top} \in \mathbb{R}^{d \times d}$ as the factorization of the low-rank adapter for the $i$th weight matrix for the $t_{th}$ task. Then the objective reduces to:

$$\min_{\boldsymbol{W}} \sum_{t=1}^{T} \min_{\{\boldsymbol{U}_i^{(t)}, \boldsymbol{V}_i^{(t)}\}_{i=1}^m} \sum_{j=1}^{n_t} \mathcal{L}\left(\Phi\left(\boldsymbol{x}_{t,j}\left(\boldsymbol{W}_1 + \boldsymbol{U}_1^{(t)} \boldsymbol{V}_1^{(t)\top}, \ldots, \boldsymbol{W}_m + \boldsymbol{U}_m^{(t)} \boldsymbol{V}_m^{(t)\top}\right)\right), \boldsymbol{y}_{t,j}\right). \tag{5}$$

In this case, we refer to the objective function as Meta-LoRA. This proposed optimization problem is designed to replace the standard retraining objective in (1). After minimizing (4) we recover base parameters $\hat{\boldsymbol{W}}_{\text{Meta}}$ that are explicitly designed to be adaptable downstream. To perform finetuning, we then run the exact same minimization in (2) but using retrained weights $\hat{\boldsymbol{W}}_{\text{Meta}}$ instead of $\hat{\boldsymbol{W}}_{\text{SR}}$.

## 3 THEORETICAL RESULTS

To establish our theoretical results, we consider $T$ multi-output linear regression retraining tasks and one test task, with the caveat that the ground-truth regressor for each task is a low-rank modification of a common single matrix. More precisely, consider the matrix $\boldsymbol{A}^* \in \mathbb{R}^{d \times d}$, which is a common parameter shared across all tasks, and task-specific adapters $\boldsymbol{U}_t^* \boldsymbol{U}_t^{*\top}$ for $t \in [T+1]$, where $\boldsymbol{U}_t^* \in \mathbb{R}^{d \times k}$ and the entries of $\boldsymbol{U}_t^*$ are i.i.d. from $\mathcal{N}(0,1)$. We work in the setting where $k \ll d$ and $k(T+1) < d$. Assume the data generation for task $\mathcal{T}_t \sim \mathcal{D}$ is given by $\boldsymbol{y}_{t,j} = (\boldsymbol{A}^* + \boldsymbol{U}_t^* \boldsymbol{U}_t^{*\top}) \boldsymbol{x}_{t,j} + \boldsymbol{\epsilon}_{t,j}$. Here, $\boldsymbol{x}_{t,j}$ is the $j$-th sample of task $t$ which is i.i.d. from $\mathcal{N}(\boldsymbol{0}, \sigma_x^2 \boldsymbol{I}_d)$, and $\boldsymbol{\epsilon}_{t,j}$ is i.i.d. $\mathcal{N}(\boldsymbol{0}, \sigma_\epsilon^2 \boldsymbol{I}_d)$ noise sampled independently of the data $\boldsymbol{x}_{t,j}$. As mentioned above, $\boldsymbol{A}^*$ can be considered as the common parameter which is close to the ground truth of each task up to a low-rank adaptation.

For each task $t$, the learner uses the linear predictor $\Phi(\boldsymbol{x}; \boldsymbol{A}_t) = \boldsymbol{A}_t \boldsymbol{x}$ for $\boldsymbol{A}_t \in \mathbb{R}^{d \times d}$, $\boldsymbol{x} \in \mathbb{R}^d$. In the ideal case, we hope to recover parameter value $\hat{\boldsymbol{A}} = \boldsymbol{A}^*$ in the retraining phase so that the fine-tuned model $\Phi_{\text{FT}}(\boldsymbol{x}; \hat{\boldsymbol{A}}, \boldsymbol{U}, \boldsymbol{V}) = (\hat{\boldsymbol{A}} + \boldsymbol{U}\boldsymbol{V}^\top)\boldsymbol{x}$ with a proper low-rank adapter $\boldsymbol{U}\boldsymbol{V}^\top$ can fit the data distribution of any downstream task also drawn from $\mathcal{D}$.

Given $N$ samples for each task, the loss for each task is $\mathcal{L}_t^N(\boldsymbol{A}_t) = \frac{1}{2N} \sum_{j=1}^N \|\boldsymbol{y}_{t,j} - \boldsymbol{A}_t \boldsymbol{x}_{t,j}\|_2^2 = \frac{1}{2N} \sum_{j=1}^N \|(\boldsymbol{A}^* + \boldsymbol{U}_t^* \boldsymbol{U}_t^{*\top} - \boldsymbol{A}_t)\boldsymbol{x}_{t,j} + \boldsymbol{\epsilon}_{t,j}\|_2^2$. We define $\mathcal{L}_t(\boldsymbol{A}_t)$ as the shifted and scaled infinite sample loss:

$$\mathcal{L}_t(\boldsymbol{A}_t) = \frac{1}{2} \left\| \boldsymbol{A}^* + \boldsymbol{U}_t^* \boldsymbol{U}_t^{*\top} - \boldsymbol{A}_t \right\|_F^2 = \frac{1}{\sigma_x^2}\left( \mathbb{E}\left[\mathcal{L}_t^N(\boldsymbol{A}_t)\right] - \frac{\sigma_\epsilon^2}{2} \right) \tag{6}$$

We assume access to infinite samples during the retraining process, as in practice, we have access to large retraining tasks relative to the low-resource downstream tasks to be used for fine-tuning.

**Remark 1.** *For convenience, we require a mild sense of task diversity and assume that the aggregated columns from all $\boldsymbol{U}_t^*$, $t \in [T+1]$, form a linearly independent set. Precisely, we assume* $\dim\left(\text{im}(\boldsymbol{U}_1^*) \oplus \cdots \oplus \text{im}(\boldsymbol{U}_{T+1}^*)\right) = k(T+1)$. *Since $k(T+1) < d$, the nature of the generation process of each $\boldsymbol{U}_t^*$ ensures that this assumption holds almost surely.*

Given access to the loss functions defined in 6, the goal of the learner is to find an $\hat{\boldsymbol{A}}$ that can be adapted to the unseen task $\mathcal{T}_{T+1}$. The infinite sample test loss for adapter factors $\boldsymbol{U}_{T+1}, \boldsymbol{V}_{T+1}$ and fixed $\hat{\boldsymbol{A}}$ is the LoRA loss on $\mathcal{T}_{T+1}$ which reduces to the low-rank matrix factorization problem:

$$\mathcal{L}_{\text{Test}}(\boldsymbol{U}_{T+1}, \boldsymbol{V}_{T+1}; \hat{\boldsymbol{A}}) = \frac{1}{2} \left\| \boldsymbol{A}^* + \boldsymbol{U}_{T+1}^* \boldsymbol{U}_{T+1}^{*\top} - \hat{\boldsymbol{A}} - \boldsymbol{U}_{T+1}\boldsymbol{V}_{T+1}^\top \right\|_F^2. \tag{7}$$

We compare the standard retraining and Meta-LoRA objectives for utilizing each $\mathcal{L}_t$ to recover a common set of base parameters $\hat{\boldsymbol{A}}$ whose low-rank adaptation $\hat{\boldsymbol{A}} + \boldsymbol{U}_{T+1}\boldsymbol{V}_{T+1}^\top$ minimizes the test loss $\mathcal{L}_{\text{Test}}$ for some $\boldsymbol{U}_{T+1}, \boldsymbol{V}_{T+1}$. We include complete proofs for all theorems in Appendix B.

### 3.1 STANDARD RETRAINING THEN FINE-TUNING

First, consider the standard retraining then fine-tuning setup as a candidate for ultimately minimizing 7. Here, the learner first finds a single matrix $\hat{\boldsymbol{A}}_{\text{SR}}$ that minimizes the sum of losses $\sum_{t=1}^T \mathcal{L}_t$:

$$\hat{\boldsymbol{A}}_{\text{SR}} = \underset{\boldsymbol{A}}{\arg\min} \; \frac{1}{2} \sum_{t=1}^T \left\| \boldsymbol{A}^* + \boldsymbol{U}_t^* \boldsymbol{U}_t^{*\top} - \boldsymbol{A} \right\|_F^2. \tag{8}$$

Then when given a new task $\mathcal{T}_{T+1}$, the learner runs LoRA to minimize the loss over the unseen task in 7. However, this strategy suffers substantial loss on the test task.

**Theorem 1.** *For standard retraining,* $\text{rank}(\hat{\boldsymbol{A}}_{SR} - \boldsymbol{A}^*) = kT$.

The above theorem demonstrates that the standard retraining process is unable to recover the ground truth shared matrix $\boldsymbol{A}^*$. Specifically, it shows that the discrepancy between the obtained solution and the ground truth $\boldsymbol{A}^*$ has a rank of $kT$. Consequently, any fine-tuning method constrained to a rank lower than $kT$ will fail to recover the correct model for the downstream task. This result follows from the fact that the obtained model from the standard retraining scheme can be written as

$$\hat{\boldsymbol{A}}_{\text{SR}} = \boldsymbol{A}^* + \frac{1}{T}\sum_{t=1}^{T} \boldsymbol{U}_t^* \boldsymbol{U}_t^{*\top}. \tag{9}$$

Now, given the fact that $\boldsymbol{U}_t^*$ are linearly independent, it follows that $\text{rank}(\sum_{t=1}^{T} \boldsymbol{U}_t^* \boldsymbol{U}_t^{*\top}) = kT$. Hence, $\hat{\boldsymbol{A}}_{\text{SR}}$ is far from both $\boldsymbol{A}^*$ and the test task ground truth parameters.

**Corollary 1.** *For number of retraining tasks $T \geq 1$, if test task adaptation rank $k' < k(T+1)$, then $\mathcal{L}_{Test}(\boldsymbol{U}_{T+1}, \boldsymbol{V}_{T+1} ; \hat{\boldsymbol{A}}_{SR}) > 0$ for all rank-$k'$ adapters $\boldsymbol{U}_{T+1}\boldsymbol{V}_{T+1}^{\top}$ where $\boldsymbol{U}_{T+1}, \boldsymbol{V}_{T+1} \in \mathbb{R}^{d\times k'}$.*

**Corollary 2.** *For a large number of retraining tasks $T$ and test task adaptation rank $k' < k(T+1)$, $\mathcal{L}_{Test}(\boldsymbol{U}_{T+1}, \boldsymbol{V}_{T+1} ; \hat{\boldsymbol{A}}_{SR}) = \Omega\left((d-k')k^2\right)$ for all $\boldsymbol{U}_{T+1}, \boldsymbol{V}_{T+1} \in \mathbb{R}^{d\times k'}$.*

Both corollaries follow from the classic result of (Mirsky, 1960). In the infinite sample setting, the LoRA rank needed to fit the test task after standard retraining is $k(T+1)$, and using anything smaller results in test error that scales with $d$. Thus, **standard retraining recovers parameters that cannot be low-rank adapted to any relevant task**. To address these issues, we employ the Meta-LoRA objective which explicitly searches for a low-rank adaptable solution.

## 3.2 META-LoRA

Although we have shown that standard retraining can lead to large losses on downstream tasks after LoRA, it is not yet clear whether any other retraining method can do better in this setting. We next explore whether minimizing the Meta-LoRA objective results in a matrix $\hat{\boldsymbol{A}}_{\text{Meta}}$ that indeed leads to a smaller test loss $\mathcal{L}_{Test}(\boldsymbol{U}_{T+1}, \boldsymbol{V}_{T+1} ; \hat{\boldsymbol{A}}_{\text{Meta}})$ for some values of $\boldsymbol{U}_{T+1}, \boldsymbol{V}_{T+1}$.

As in (5), we introduce low-rank adapters during the retraining phase to model the different training tasks. We search for a value of $\boldsymbol{A}$ such for all $\mathcal{T}_t$, the loss $\mathcal{L}_t$ after running LoRA on $\mathcal{T}_t$ is minimized. This promotes values of $\boldsymbol{A}$ that can be easily adapted to unseen tasks downstream. We use the Meta-LoRA loss but with symmetric low-rank adapters $\boldsymbol{U}_t\boldsymbol{U}_t^{\top}$ for the $t_{th}$ task $\mathcal{T}_t$ in retraining. We allow asymmetric adapters at test time. The infinite sample Meta-LoRA loss is then

$$\mathcal{L}_{\text{Meta}}(\boldsymbol{A}) = \sum_{t=1}^{T} \min_{\boldsymbol{U}_t} \mathcal{L}_t(\boldsymbol{A} + \boldsymbol{U}_t\boldsymbol{U}_t^{\top}). \tag{10}$$

Define the concatenation of each $\boldsymbol{U}_t$ as $\boldsymbol{U} = (\boldsymbol{U}_1, \dots, \boldsymbol{U}_T) \in (\mathbb{R}^{d\times k})^T$. Then minimizing (10) is equivalent to solving $\min_{\boldsymbol{A},\boldsymbol{U}} \mathcal{L}(\boldsymbol{A},\boldsymbol{U})$ where

$$\mathcal{L}(\boldsymbol{A},\boldsymbol{U}) = \frac{1}{2}\sum_{t=1}^{T} \left\| \boldsymbol{A}^* + \boldsymbol{U}_t^*\boldsymbol{U}_t^{*\top} - \boldsymbol{A} - \boldsymbol{U}_t\boldsymbol{U}_t^{\top} \right\|_F^2. \tag{11}$$

We have seen that standard retraining does not recover an optimal solution, but it is unclear what the global minima of this new objective function are and if they can be easily found. Note that by fixing $\boldsymbol{A}$, (11) is $T$ independent symmetric matrix factorization problems, and by fixing $\boldsymbol{U}$, (11) is a convex quadratic problem over $\boldsymbol{A}$. Despite these well-understood sub-problems, joint minimization over $\boldsymbol{A}$ and $\boldsymbol{U}$ presents challenging variable interactions that complicate the analysis. Nevertheless, we employ a careful landscape analysis of (11) to address these questions.

### 3.2.1 LANDSCAPE OF GLOBAL MINIMA OF (11)

First, we show that the objective is well-posed, i.e., minimization of $\mathcal{L}$ leads to an adaptable solution.

**Theorem 2.** *For any $T \geq 1$, if $\mathcal{L}(\hat{\boldsymbol{A}}, \hat{\boldsymbol{U}}) = 0$, then $\hat{\boldsymbol{A}} = \boldsymbol{A}^* + \boldsymbol{C}$ where $\text{rank}(\boldsymbol{C}) \leq 2k$*

Clearly, any point is a global minimum of (11) if and only if it achieves zero loss. Theorem 2 guarantees that the values of $\boldsymbol{A}$ that induce global minima of (11) are at most rank-$2k$ away from the ground truth parameter $\boldsymbol{A}^*$.

**Corollary 3.** *For any $T \geq 1$, if $\mathcal{L}(\hat{A}, \hat{U}) = 0$, then there exists a rank-$3k$ adapter $U_{T+1} V_{T+1}^\top$ where $U_{T+1}, V_{T+1} \in \mathbb{R}^{d \times 3k}$ such that $\mathcal{L}_{Test}(U_{T+1}, V_{T+1} ; \hat{A}) = 0$.*

This again follows from classic low-rank factorization results, as $\mathcal{L}_{Test}(U_{T+1}, V_{T+1} ; A^* + C) = \frac{1}{2} \left\| U_{T+1}^* U_{T+1}^{*\top} - C - U_{T+1} V_{T+1}^\top \right\|_F^2$ and $\text{rank}\left(U_{T+1}^* U_{T+1}^{*\top} - C\right) \leq 3k$. Note that $3k$ is still much smaller than $d$ as $k \ll d$.

*Proof sketch of Theorem 2.* Notice that any set of parameters $(\hat{A}, \hat{U})$ such that $\mathcal{L}(\hat{A}, \hat{U}) = 0$ must be a critical point as $\mathcal{L} \geq 0$. This directly implies that $\hat{A} = A^* + \frac{1}{T} \sum_{t=1}^T U_t^* U_t^{*\top} - U_t U_t^\top$ and $U_t^* U_t^{*\top} - U_t U_t^\top = U_j^* U_j^{*\top} - U_j U_j^\top$ for all $1 \leq i, j \leq T$. It then follows that $\hat{A} = A^* + U_1^* U_1^{*\top} - U_1 U_1^\top$, and $\text{rank}(U_1^* U_1^{*\top} - U_1 U_1^\top) \leq \text{rank}(U_1^* U_1^{*\top}) + \text{rank}(U_1 U_1^\top) \leq 2k$. □

This result shows that for any $T \geq 1$, any global minimum of (11) recovers $A^*$ with an error up to rank-$2k$. Consequently, it can perform well on a downstream task after fine-tuning with a rank-$3k$ adaptor. Furthermore, we demonstrate that when the number of tasks satisfies $T \geq 3$, a stronger result can be established. Specifically, in this case, we can prove the exact recovery of the ground truth parameter $A^*$ is possible.

**Theorem 3.** *If $T \geq 3$, then $\mathcal{L}(\hat{A}, \hat{U}) = 0$ implies $\hat{A} = A^*$ and $U_t U_t^\top = U_t^* U_t^{*\top}$ for all $t \in [T]$*

Theorem 3 guarantees that the ground truth parameters are the unique global minimum up to orthogonal symmetry when there are three or more tasks, regardless of the ambient dimension or the number of columns $k$. This result is surprising, as most theoretical results for multi-task learning require higher task diversity, typically where the number of tasks $T$ is required to be larger than the effective task dimension $k$ (Du et al., 2021; Collins et al., 2022). However, we establish this uniqueness result for the absolute condition $T \geq 3$. This implies that exact test task fine-tuning can be achieved with a rank $k$-adaptation.

**Corollary 4.** *For any $T \geq 3$, if $\mathcal{L}(\hat{A}, \hat{U}) = 0$, then there exists a rank-$k$ adapter $U_{T+1} V_{T+1}^\top$ where $U_{T+1}, V_{T+1} \in \mathbb{R}^{d \times k}$ such that $\mathcal{L}_{Test}(U_{T+1}, V_{T+1} ; \hat{A}) = 0$.*

This follows directly from $\mathcal{L}_{Test}(U_{T+1}, V_{T+1} ; A^*) = \frac{1}{2} \left\| U_{T+1}^* U_{T+1}^{*\top} - U_{T+1} V_{T+1}^\top \right\|_F^2$ and $\text{rank}\left(U_{T+1}^* U_{T+1}^{*\top}\right) = k$.

*Proof sketch of Theorem 3.* We again rely on the fact that a set of parameters that achieves zero loss must satisfy $U_t^* U_t^{*\top} - \hat{U}_t \hat{U}_t^\top = U_s^* U_s^{*\top} - \hat{U}_s \hat{U}_s^\top$ for all $t, s \in [T]$. Then

$$U_1^* U_1^{*T} = \hat{U}_1 \hat{U}_1^T + U_2^* U_2^{*T} - \hat{U}_2 \hat{U}_2^T = \hat{U}_1 \hat{U}_1^T + U_3^* U_3^{*T} - \hat{U}_3 \hat{U}_3^T.$$

Since $U_1^* U_1^{*T} \succcurlyeq 0$, both $\text{im}(\hat{U}_2) \subseteq \text{im}(\hat{U}_1) + \text{im}(U_2^*)$ and $\text{im}(\hat{U}_3) \subseteq \text{im}(\hat{U}_1) + \text{im}(U_3^*)$. This then implies that the image of $\hat{U}_1$ is a subset of two key subspaces:

$$\text{im}(U_1^*) \subseteq \text{im}(\hat{U}_1) + \text{im}(U_2^*) \text{ and } \text{im}(U_1^*) \subseteq \text{im}(\hat{U}_1) + \text{im}(U_3^*). \tag{12}$$

We then make use of a key lemma to prove the result. The proof can be found in Appendix B.3.

**Lemma 1.** $([\text{im}(\hat{U}_1) \oplus \text{im}(U_2^*)] \cap [\text{im}(\hat{U}_1) \oplus \text{im}(U_3^*)]) = \text{im}(\hat{U}_1)$

Combining Lemma 1 with (12) implies that $\text{im}(\hat{U}_1) = \text{im}(U_1^*)$. Then applying the same argument for the other indices shows that $\text{im}(\hat{U}_t) = \text{im}(U_t^*)$ for all $t \in [T]$. Since $U_t^* U_t^{*\top} - \hat{U}_t \hat{U}_t^\top = U_s^* U_s^{*\top} - \hat{U}_s \hat{U}_s^\top$ for all $t, s \in [T]$ and $\text{im}(U_1^*) \cap \text{im}(U_2^*) = \{0\}$, it follows that $U_t^* U_t^{*\top} = \hat{U}_t \hat{U}_t^\top$ for all $t \in [T]$. Then since $\nabla_A \mathcal{L}(\hat{A}, \hat{U}) = 0$, $\hat{A} = A^* + \frac{1}{T} \sum_{t=1}^T U_t^* U_t^{*\top} - \hat{U}_t \hat{U}_t^\top = A^*$. □

The proof of Theorem 3 relies on the assumption that there are at least three training tasks. This is necessary to some degree as if there are only two tasks, we can construct ground truth parameters that have infinite solutions as in the example in Appendix D.1.

**Summary.** The previous two theorems show that for any $T \geq 1$, the set of global minima of the meta objective is always adaptable to the downstream task. Furthermore, if $T \geq 3$, the global minima of

the meta-objective are the unique ground truth parameters $(\boldsymbol{A}^*, \boldsymbol{U}_t^*)$ up to orthogonal symmetry of $\boldsymbol{U}_t$. In other words, minimizing (11) guarantees the recovery of the ground truth parameters.

### 3.2.2 Algorithms for Minimizing (11)

The above results establish that minimizing the meta-objective (11) leads to recovery of the ground truth parameters, with a small error term when $T = 2$. However, it is unclear if this minimization problem can always be solved by local optimization methods.

**Theorem 4.** *If $T = 2$, then $\mathcal{L}(\hat{\boldsymbol{A}}, \hat{\boldsymbol{U}}) = 0$ if and only if $(\hat{\boldsymbol{A}}, \hat{\boldsymbol{U}})$ is a second order stationary point (SOSP) of $\mathcal{L}$.*

Thus, local optimization algorithms for finding SOSPs, such as perturbed gradient descent and cubic-regularized Newton method, can efficiently find the minima of the meta-learning objective.

*Proof sketch.* Clearly if $\mathcal{L}(\hat{\boldsymbol{A}}, \hat{\boldsymbol{U}}) = 0$, then $(\hat{\boldsymbol{A}}, \hat{\boldsymbol{U}})$ is an SOSP. The reverse direction is the challenging part of the proof. We equivalently prove that if $(\hat{\boldsymbol{A}}, \hat{\boldsymbol{U}})$ is a critical point and $\mathcal{L}(\hat{\boldsymbol{A}}, \hat{\boldsymbol{U}}) \neq 0$, then $\nabla^2 \mathcal{L}(\hat{\boldsymbol{A}}, \hat{\boldsymbol{U}})$ has a negative eigenvalue.

Assume for the sake of contradiction that $(\hat{\boldsymbol{A}}, \hat{\boldsymbol{U}})$ is an SOSP and $\mathcal{L}(\hat{\boldsymbol{A}}, \hat{\boldsymbol{U}}) \neq 0$. Considering $\mathcal{L}$ as a function of the flattened vector $[\text{vec}(\boldsymbol{A}); \text{vec}(\boldsymbol{U}_1); \text{vec}(\boldsymbol{U}_2)]$, the idea of the proof is to contradict the assumption that $\nabla^2 \mathcal{L} \succeq \boldsymbol{0}$.

Since $\nabla_{\boldsymbol{A}}^2 \mathcal{L}(\hat{\boldsymbol{A}}, \hat{\boldsymbol{U}}) = T\boldsymbol{I} \succ \boldsymbol{0}$, we can work with the Schur complement $\boldsymbol{Q} = \left( \nabla^2 \mathcal{L} / \nabla_{\boldsymbol{A}}^2 \mathcal{L} \right)(\hat{\boldsymbol{A}}, \hat{\boldsymbol{U}})$ as $\nabla^2 \mathcal{L}(\hat{\boldsymbol{A}}, \hat{\boldsymbol{U}}) \succeq \boldsymbol{0}$ if and only if $\boldsymbol{Q} \succeq \boldsymbol{0}$. Inspection of the condition $\nabla \mathcal{L}(\hat{\boldsymbol{A}}, \hat{\boldsymbol{U}}) = \boldsymbol{0}$ along with the assumptions $\mathcal{L}(\hat{\boldsymbol{A}}, \hat{\boldsymbol{U}}) \neq 0$ and $T = 2$ gives three key properties:

$$\left( \hat{\boldsymbol{U}}_2 \hat{\boldsymbol{U}}_2^\top - \hat{\boldsymbol{U}}_1 \hat{\boldsymbol{U}}_1^\top \right) \boldsymbol{x} = \left( \boldsymbol{U}_2^* \boldsymbol{U}_2^{*\top} - \boldsymbol{U}_1^* \boldsymbol{U}_1^{*\top} \right) \boldsymbol{x} \quad \forall \boldsymbol{x} \text{ s.t. } \left( \hat{\boldsymbol{U}}_2 \hat{\boldsymbol{U}}_2^\top - \hat{\boldsymbol{U}}_1 \hat{\boldsymbol{U}}_1^\top \right) \boldsymbol{x} \neq \boldsymbol{0} \quad (13)$$

$$\hat{\boldsymbol{U}}_2 \hat{\boldsymbol{U}}_2^\top - \hat{\boldsymbol{U}}_1 \hat{\boldsymbol{U}}_1^\top \neq \boldsymbol{U}_2^* \boldsymbol{U}_2^{*\top} - \boldsymbol{U}_1^* \boldsymbol{U}_1^{*\top} \quad (14)$$

$$\dim \ker \left( \hat{\boldsymbol{U}}_2 \hat{\boldsymbol{U}}_2^\top - \hat{\boldsymbol{U}}_1 \hat{\boldsymbol{U}}_1^\top \right) > 0 \quad (15)$$

Thus, there is an eigenvector $\boldsymbol{z}$ of $\boldsymbol{U}_2^* \boldsymbol{U}_2^{*\top} - \boldsymbol{U}_1^* \boldsymbol{U}_1^{*\top}$ with eigenvalue $\lambda \neq 0$ such that $\boldsymbol{z} \in \ker(\hat{\boldsymbol{U}}_2 \hat{\boldsymbol{U}}_2^\top - \hat{\boldsymbol{U}}_1 \hat{\boldsymbol{U}}_1^\top)$. Assume without loss of generality $\lambda > 0$, and consider $\boldsymbol{\alpha} \in \mathbb{R}^{2k}$. Define $g(\cdot ; \boldsymbol{z}) : \mathbb{R}^{2k} \to \mathbb{R}$ such that $g(\boldsymbol{\alpha}; \boldsymbol{z}) = (\boldsymbol{\alpha} \otimes \boldsymbol{z})^\top \boldsymbol{Q} (\boldsymbol{\alpha} \otimes \boldsymbol{z})$, where $\boldsymbol{\alpha} = [\boldsymbol{\alpha}_1; \boldsymbol{\alpha}_2]$ with $\boldsymbol{\alpha}_1, \boldsymbol{\alpha}_2 \in \mathbb{R}^k$. Then,

$$g(\boldsymbol{\alpha}; \boldsymbol{z}) = \left\| \hat{\boldsymbol{U}}_1 \boldsymbol{\alpha}_1 + \hat{\boldsymbol{U}}_2 \boldsymbol{\alpha}_2 \right\|_2^2 + \lambda \left( \|\boldsymbol{\alpha}_1\|_2^2 - \|\boldsymbol{\alpha}_2\|_2^2 \right). \quad (16)$$

We prove the existence of $\boldsymbol{\alpha} \in \mathbb{R}^{2k}, \boldsymbol{x} \in \mathbb{R}^d$ such that $g(\boldsymbol{\alpha}; \boldsymbol{x}) < 0$ considering two different cases. Define $N^- : S_d \to \mathbb{Z}$ as the function that returns the number of negative eigenvalues of its input.

**Case 1**: $N^-(\hat{\boldsymbol{U}}_2 \hat{\boldsymbol{U}}_2^\top - \hat{\boldsymbol{U}}_1 \hat{\boldsymbol{U}}_1^\top) < k$: Then there exists $\boldsymbol{z}^- \in \mathbb{R}^d$ that is a $\lambda^-$-eigenvector of $\boldsymbol{U}_2^* \boldsymbol{U}_2^{*T} - \boldsymbol{U}_1^* \boldsymbol{U}_1^{*T}, \lambda^- < 0$, where $\boldsymbol{z} \in \ker \left( \hat{\boldsymbol{U}}_2 \hat{\boldsymbol{U}}_2^\top - \hat{\boldsymbol{U}}_1 \hat{\boldsymbol{U}}_1^\top \right)$. By (15), we can pick $\boldsymbol{\alpha}$ such that $\hat{\boldsymbol{U}}_1 \boldsymbol{\alpha}_1 + \hat{\boldsymbol{U}}_2 \boldsymbol{\alpha}_2 = 0, \boldsymbol{\alpha}_1, \boldsymbol{\alpha}_2 \neq 0$. Then $g(\boldsymbol{\alpha}; \boldsymbol{z}) = -g(\boldsymbol{\alpha}; \boldsymbol{z}^-) = \|\boldsymbol{\alpha}_1\|_2^2 - \|\boldsymbol{\alpha}_2\|_2^2$.

If $\|\bar{\boldsymbol{\alpha}}_1\|_2 \neq \|\bar{\boldsymbol{\alpha}}_2\|_2$, $\min\{g(\boldsymbol{\alpha}; \boldsymbol{z}), g(\boldsymbol{\alpha}; \boldsymbol{z}^-)\} < 0$. Else, $g(\bar{\boldsymbol{\alpha}}; \boldsymbol{z}) = 0$, but $\nabla_{\boldsymbol{\alpha}_1} g(\bar{\boldsymbol{\alpha}}; \boldsymbol{z}) = \hat{\boldsymbol{U}}_1^\top (\hat{\boldsymbol{U}}_1 \bar{\boldsymbol{\alpha}}_1 + \hat{\boldsymbol{U}}_2 \bar{\boldsymbol{\alpha}}_2) - 2\lambda \bar{\boldsymbol{\alpha}}_2 = -2\lambda \bar{\boldsymbol{\alpha}}_2 \neq 0$. Thus there exists $\bar{\boldsymbol{\alpha}}$ in an infinitesimal neighborhood around $\boldsymbol{\alpha}$ where $g(\bar{\boldsymbol{\alpha}} ; \boldsymbol{z}) < 0$.

**Case 2**: $N^-(\hat{\boldsymbol{U}}_2 \hat{\boldsymbol{U}}_2^\top - \hat{\boldsymbol{U}}_1 \hat{\boldsymbol{U}}_1^\top) = k$: By (15), $\exists \boldsymbol{\Gamma} \in O_k$ such that $\hat{\boldsymbol{U}}_2 \boldsymbol{\Gamma} \boldsymbol{e}_1 \in (\text{im}(\hat{\boldsymbol{U}}_1) \cap \text{im}(\hat{\boldsymbol{U}}_2))$. Define $\boldsymbol{y} = \hat{\boldsymbol{U}}_2 \boldsymbol{\Gamma} \boldsymbol{e}_1$. Then

$$k = N^-(-\hat{\boldsymbol{U}}_1 \hat{\boldsymbol{U}}_1^\top) \geq N^-(\boldsymbol{y}\boldsymbol{y}^\top - \hat{\boldsymbol{U}}_1 \hat{\boldsymbol{U}}_1^\top) \geq N^-(\hat{\boldsymbol{U}}_2 \hat{\boldsymbol{U}}_2^\top - \hat{\boldsymbol{U}}_1 \hat{\boldsymbol{U}}_1^\top) = k.$$

Thus, $N^-(\boldsymbol{y}\boldsymbol{y}^\top - \hat{\boldsymbol{U}}_1 \hat{\boldsymbol{U}}_1^\top) = k$ and $\text{rank}(\boldsymbol{y}\boldsymbol{y}^\top - \hat{\boldsymbol{U}}_1 \hat{\boldsymbol{U}}_1^\top) \leq k$, so $\boldsymbol{y}\boldsymbol{y}^\top - \hat{\boldsymbol{U}}_1 \hat{\boldsymbol{U}}_1^\top \preccurlyeq 0$. Take $\boldsymbol{\alpha}$ such that $\hat{\boldsymbol{U}}_1 \boldsymbol{\alpha}_1 = -\boldsymbol{y}$ and $\boldsymbol{\alpha}_2 = \boldsymbol{\Gamma} \boldsymbol{e}_1$. Then $\boldsymbol{y}_1 \boldsymbol{y}_1^\top - \hat{\boldsymbol{U}}_1 \hat{\boldsymbol{U}}_1^\top = \hat{\boldsymbol{U}}_1 (\boldsymbol{\alpha}_1 \boldsymbol{\alpha}_1^\top - \boldsymbol{I}) \hat{\boldsymbol{U}}_1^\top \preccurlyeq 0$. Therefore $\|\boldsymbol{\alpha}_1\|_2 \leq 1$. Then $g(\boldsymbol{\alpha}; \boldsymbol{z}) = \|\hat{\boldsymbol{U}}_1 \boldsymbol{\alpha}_1 + \hat{\boldsymbol{U}}_2 \boldsymbol{\alpha}_2\|_2^2 + \lambda(\|\boldsymbol{\alpha}_1\|_2^2 - \|\boldsymbol{\alpha}_2\|_2^2) = \lambda(\|\boldsymbol{\alpha}_1\|_2^2 - 1) \leq 0$.

If $g(\boldsymbol{\alpha}; \boldsymbol{z}) < 0$ we are done. Else, the same analysis from Case 1 will show that $\nabla g(\bar{\boldsymbol{\alpha}}; \boldsymbol{z}) \neq \boldsymbol{0}$, so there exists $\bar{\boldsymbol{\alpha}}$ in an infinitesimal neighborhood around $\boldsymbol{\alpha}$ where $g(\bar{\boldsymbol{\alpha}} ; \boldsymbol{z}) < 0$. □

**Summary.** We have shown that when $T = 2$, any optimization algorithm for finding an SOSP will find a global minimum of the meta-objective (11). Surprisingly, when there are three or more tasks, numerical experiments (see Appendix D.2) show that adversarially picking $\boldsymbol{U}_t^*$ can result in specific instantiations of (11) with spurious local minima. In the next section, we perform extensive numerical experiments for various values of $T$ which show that these spurious minima are almost never found in practice and vanilla gradient descent is sufficient to minimize (11).

## 4 EXPERIMENTS

### 4.1 LINEAR EXPERIMENTS

To test our algorithm, we perform experiments on a synthetic dataset. We generate $\boldsymbol{A}^* \in \mathbb{R}^{d \times d}$ and $\boldsymbol{U}_t^* \in \boldsymbol{R}^{d \times k}$ for all tasks $t \in [T + 1]$, where the entries of $\boldsymbol{A}^*$ and each $\boldsymbol{U}_t^*$ are i.i.d. $\mathcal{N}(0, 1)$ random variables. Then we generate $\frac{N}{T}$ samples for each retraining task $t \in [T]$ as $\boldsymbol{y}_{t,j} = (\boldsymbol{A}^* + \boldsymbol{U}_t^* \boldsymbol{U}_t^{*\top}) \boldsymbol{x}_{t,j} + \boldsymbol{\epsilon}_{t,j}, j \in [\frac{N}{T}]$ and $N'$ samples for the held-out task as as $\boldsymbol{y}_{T+1,j} = (\boldsymbol{A}^* + \boldsymbol{U}_{T+1}^* \boldsymbol{U}_{T+1}^{*\top}) \boldsymbol{x}_{T+1,j} + \boldsymbol{\epsilon}_{T+1,j}, j \in [N']$, where $\boldsymbol{x}_{t,j} \sim \mathcal{N}(0, \boldsymbol{I}_d)$ and $\boldsymbol{\epsilon}_{t,j} \sim \mathcal{N}(0, \sigma_\epsilon^2 \boldsymbol{I}_d)$ are i.i.d. feature and noise vectors respectively.

We apply gradient descent to the Meta-LoRA and standard retraining objectives on the $T$ retraining tasks and then fine-tune to the $(T+1)$-th task using LoRA. We use symmetric adapters for the Meta-LoRA retraining objective and asymmetric adapters during fine-tuning for each retraining method. We conduct experiments by varying one hyperparameter at a time from the fixed values of $d = 10, T = 3, N = 5000, N' = 100, k = 1$ and $\sigma = 0.1$. When $T = 2$, we use a rank-3 adaptation during fine-tuning and use a rank-1 adaptation otherwise for both retraining schemes.

We plot the population loss on the test task after training and fine-tuning with Meta-LoRA and SR+LoRA, respectively, in Figure 1. Meta-LoRA significantly outperforms SR+LoRA for all data generation parameter settings. We observe from Figure 1b that with more retraining data, Meta-LoRA performance first improves and then stagnates because of the finite sample noise floor during the fine-tuning stage. We observe a similar phenomenon in Figure 1c. Figure 1d shows that the performance of Meta-LoRA improves for $T > 2$ relative to $T = 2$ but is agnostic to $T$ once in the $T > 2$ regime. Lastly, Figure 1a shows how performance worsens with increasing dimension.

### 4.2 LLM EXPERIMENTS

To test the Meta-LoRA objective beyond linear models, we perform experiments using the pre-trained 355 million parameter RoBERTa-Large model on the ConvAI2 dataset. ConvAI2 consists of conversations between two personas, i.e. people with different personalities. Each persona is associated with a short list of factual information that guides the content of their responses. We model learning the dialogue continuations of each individual persona as a different task. A training sample for a given persona consists of the previous conversation as input and 20 candidate dialogue continuations, where one of the 20 candidates is the true continuation. We consider the supervised learning task of selecting the correct continuation. During training, we maximize the log-likelihood of the correct continuation and minimize the log-likelihood of each of the incorrect continuations conditioned on the observed conversation history. To run inference given the past conversation and the 20 possible continuations, we select the continuation with the highest conditional likelihood.

For both the standard retraining and Meta-LoRA objectives, we retrain the model using the $T = 10$ largest retraining tasks, with an average of 117.4 training samples and 36.5 heldout samples per retraining task. We select the model from the epoch with the best average accuracy on the heldout samples and then fine-tune to each of the 10 largest test tasks. For each test task, we take the accuracy on the heldout data from the best performing epoch. We run 5 random trials for this entire retraining and fine-tuning process and report the median best heldout accuracy for each task. All training was done on a single Nvidia A40 GPU, and we report our training hyperparameters in Appendix C.

We compare performance across the test tasks in Table 1. We first minimize the Meta-LoRA objective using rank-8 adapters on the retraining tasks and denote this model Meta-LoRA-8. In Table 1a,

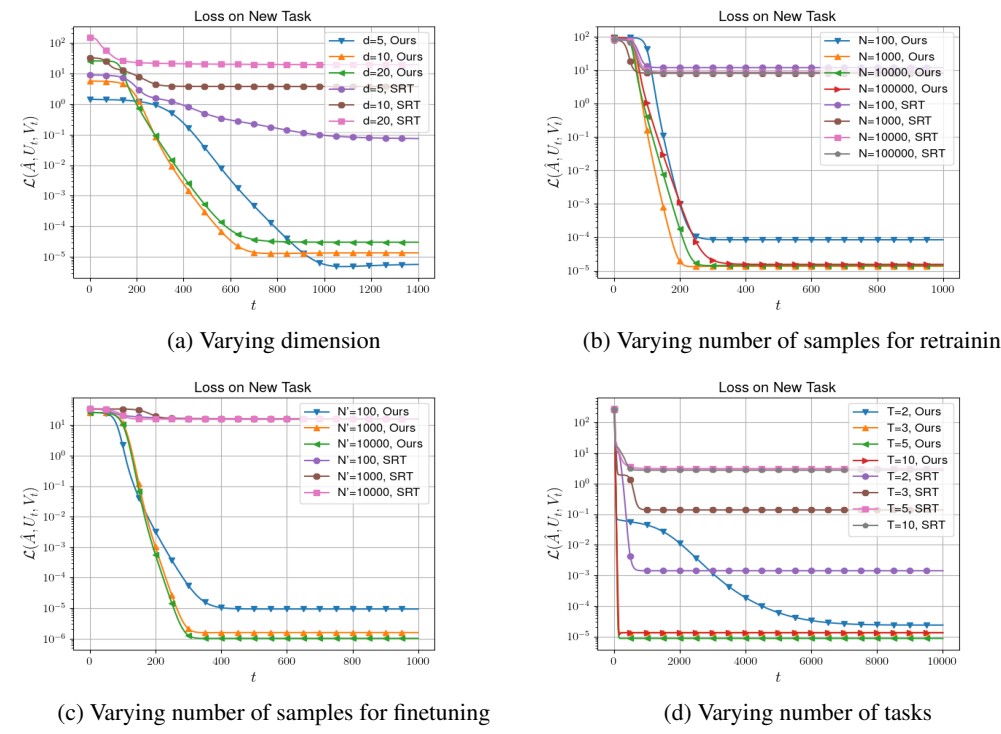

(a) Varying dimension

(b) Varying number of samples for retraining

(c) Varying number of samples for finetuning

(d) Varying number of tasks

Figure 1: Evaluating the linear Meta-LoRA algorithm in different settings.

we show this improves performance over standard retraining followed by rank-8 LoRA on the test test, denoted SR+LoRA. As suggested by Theorem 2, we test if we can improve performance by increasing the LoRA rank during fine-tuning relative to the rank of the adapters in retraining with Meta-LoRA. Table 1b shows that the retrained Meta-LoRA-8 model fine-tuned with rank-16 adaptations outperforms both standard retraining followed by rank-16 LoRA as well as the Meta-LoRA-16 model which was retrained and fine-tuned with rank-16 adaptations.

Table 1: Comparison of Meta-LoRA and the SRT+LoRA algorithms on the ConvAI2 dataset

(a) Rank-8 fine-tuning adaptations

| Algorithm | Task 1 | Task 2 | Task 3 | Task 4 | Task 5 | Task 6 | Task 7 | Task 8 | Task 9 | Task 10 | Average |
|---|---|---|---|---|---|---|---|---|---|---|---|
| SR+LoRA | 43.75 | 40.00 | 43.48 | 41.94 | 41.03 | 37.23 | 42.73 | 43.20 | 41.13 | 40.76 | 41.52 |
| Meta-LoRA-8 | **50.00** | **50.00** | **47.82** | **48.39** | **46.15** | **41.49** | **44.55** | **44.00** | **42.55** | **42.68** | **45.76** |

(b) Rank-16 fine-tuning adaptations

| Algorithm | Task 1 | Task 2 | Task 3 | Task 4 | Task 5 | Task 6 | Task 7 | Task 8 | Task 9 | Task 10 | Average |
|---|---|---|---|---|---|---|---|---|---|---|---|
| SR+LoRA | 43.75 | 43.33 | 39.13 | 38.71 | 39.74 | 35.11 | 38.18 | 39.20 | 39.72 | 38.85 | 39.57 |
| Meta-LoRA-8 | **50.0** | **53.33** | **50.0** | **50.0** | **48.72** | **42.55** | **45.45** | **44.80** | **45.39** | **44.59** | **47.48** |
| Meta-LoRA-16 | 43.75 | 33.33 | 36.96 | 40.32 | 43.59 | 39.36 | 42.73 | 41.60 | 40.43 | 40.13 | 40.22 |

## 5 CONCLUSION

We introduced the Meta-Adapters objective function for retraining an FM on a collection of tasks in a way that prepares the model for subsequent downstream fine-tuning. We provide theoretical justifications on the shortcomings of standard retraining as well as where the Meta-Adapters objective using LoRA (Meta-LoRA) can provably improve performance. Empirically, our Meta-LoRA objective outperforms standard retraining for adapting to unseen downstream tasks. Future avenues include extending our theoretical analysis to finite sample settings and to more general adapters.

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

## A   RELATED WORK ON LoRA-STYLE PEFT

There is a vast amount of work in developing PEFT methods for FMs. The LoRA algorithm (Hu et al., 2021) has established itself as a popular and successful PEFT strategy and has inspired various extensions such as QLoRA, DoRA, and others (Dettmers et al., 2023; Liu et al., 2024; Zhang et al., 2023). These algorithms are heuristics for mimicking the full finetuning of an FM to a specific downstream task and have proven to be empirically successful in various settings. However, there is a lack of theoretical analysis on the adaptability of PFMs under LoRA-style adaptations, the ability to efficiently optimize LoRA-style objectives, and the kinds of solutions they recover. Some recent works have attempted to analyze different parts of these theoretical questions.

**Convergence of LoRA.** (Jang et al., 2024) analyzes the optimization landscape for LoRA for the Neural Tangent Kernel regime. The authors show that LoRA finetuning converges in this setting as they prove that the objective function satisfies a strict saddle property, ensuring that there are no spurious local minima. However, this focuses on the actual ability of LoRA to converge to the optimal low-rank adapter given an FM, and does not consider the adaptability of the FM in the first place.

**Expressivity of LoRA.** (Zeng & Lee, 2023) derives the expressive power of LoRA as a function of model depth. This work shows that under some mild conditions, fully connected and transformer networks when respectively adapted with LoRA can closely approximate arbitrary smaller networks. They quantify the required LoRA rank to achieve this approximation as well as the resulting approximation error.

## B   PROOFS

### B.1   PROOF OF THEOREM 1 AND COROLLARIES 1,2

By definition,

$$\hat{\boldsymbol{A}}_{\text{SR}} = \arg\min_{\boldsymbol{A}} \frac{1}{2} \sum_{t=1}^{T} \left\| \boldsymbol{A}^* + \boldsymbol{U}_t^* \boldsymbol{U}_t^{*\top} - \boldsymbol{A} \right\|_F^2$$

This optimization problem is just a quadratic function of $\boldsymbol{A}$, so we can simply solve for the point at which the gradient is $\boldsymbol{0}$. Thus, $\hat{\boldsymbol{A}}_{\text{SR}}$ must satisfy:

$$\sum_{t=1}^{T} \left( \boldsymbol{A}^* + \boldsymbol{U}_t^* \boldsymbol{U}_t^{*\top} - \hat{\boldsymbol{A}}_{\text{SR}} \right) = \boldsymbol{0}$$

Thus,

$$\hat{\boldsymbol{A}}_{\text{SR}} = \boldsymbol{A}^* + \frac{1}{T} \sum_{t=1}^{T} \boldsymbol{U}_t^* \boldsymbol{U}_t^{*\top}$$

Therefore, $\text{rank}\left( \hat{\boldsymbol{A}}_{\text{SR}} - \boldsymbol{A}^* \right) = \text{rank}\left( \frac{1}{T} \sum_{t=1}^{T} \boldsymbol{U}_t^* \boldsymbol{U}_t^{*\top} \right) = kT$. Further,

$$\begin{aligned}
\mathcal{L}_{\text{Test}}(\boldsymbol{U}_{T+1}, \boldsymbol{V}_{T+1} ; \hat{\boldsymbol{A}}_{\text{SR}}) &= \frac{1}{2} \left\| \boldsymbol{A}^* + \boldsymbol{U}_{T+1}^* \boldsymbol{U}_{T+1}^{*\top} - \hat{\boldsymbol{A}}_{\text{SR}} - \boldsymbol{U}_{T+1} \boldsymbol{V}_{T+1}^{\top} \right\|_F^2 \\
&= \frac{1}{2} \left\| \boldsymbol{U}_{T+1}^* \boldsymbol{U}_{T+1}^{*\top} - \frac{1}{T} \sum_{t=1}^{T} \boldsymbol{U}_t^* \boldsymbol{U}_t^{*\top} - \boldsymbol{U}_{T+1} \boldsymbol{V}_{T+1}^{\top} \right\|_F^2 \\
&\approx \frac{1}{2} \left\| \boldsymbol{U}_{T+1}^* \boldsymbol{U}_{T+1}^{*\top} - k\boldsymbol{I} - \boldsymbol{U}_{T+1} \boldsymbol{V}_{T+1}^{\top} \right\|_F^2 \quad \text{for large } T
\end{aligned}$$

$\boldsymbol{U}_{T+1}^* \boldsymbol{U}_{T+1}^{*\top} - k\boldsymbol{I}$ has $d - k$ eigenvalues of magnitude $k$, and the rank-$k'$ factorization $\boldsymbol{U}_{T+1} \boldsymbol{V}_{T+1}^{\top}$ can only capture $k'$ of them, so $\boldsymbol{U}_{T+1}^* \boldsymbol{U}_{T+1}^{*\top} - k\boldsymbol{I} - \boldsymbol{U}_{T+1} \boldsymbol{V}_{T+1}^{\top}$ has at least $d - k' - k$ eigenvalues of magnitude $k$. Thus, $\mathcal{L}_{\text{Test}}(\boldsymbol{U}_{T+1}, \boldsymbol{V}_{T+1} ; \hat{\boldsymbol{A}}_{\text{SR}})$ scales as $(d - k' - k)k^2 \approx (d - k')k^2$ since $k \ll d$.

## B.2  PROOF OF THEOREM 2

*Proof.* Since $\mathcal{L}(\hat{A}, \hat{U}) = 0$ and $\mathcal{L} \geq 0$ we must have that $\nabla_A \mathcal{L} = 0$.

Thus, $\hat{A} = A^* - \frac{1}{T} \sum_{j=1}^{T} \left( \hat{U}_j \hat{U}_j^\top - U_j^* U_j^{*\top} \right)$. Plugging this into $\mathcal{L}$ gives

$$0 = \mathcal{L}(\hat{A}, \hat{U}) = \frac{1}{2} \sum_{t=1}^{T} \left\| A^* + U_t^* U_t^{*\top} - \left( A^* - \frac{1}{T} \sum_{s=1}^{T} \left( \hat{U}_s \hat{U}_s^\top - U_s^* U_s^{*\top} \right) \right) - U_t U_t^\top \right\|_F^2$$

$$= \frac{1}{2} \sum_{t=1}^{T} \left\| U_t^* U_t^{*\top} - U_t U_t^\top - \frac{1}{T} \sum_{s=1}^{T} \left( \hat{U}_s \hat{U}_s^\top - U_s^* U_s^{*\top} \right) \right\|_F^2.$$

Thus each term of the summation is zero, so for all $t, s \in [T]$,

$$\hat{U}_t \hat{U}_t^T - U_t^* U_t^{*T} = \hat{U}_s \hat{U}_s^T - U_s^* U_s^{*T}.$$

Combining these results gives that

$$\hat{A} = A^* - \frac{1}{T} \sum_{s=1}^{T} \left( \hat{U}_s \hat{U}_s^\top - U_s^* U_s^{*\top} \right)$$

$$= A^* - \left( \hat{U}_1 \hat{U}_1^\top - U_1^* U_1^{*\top} \right)$$

Let $C = -\left( \hat{U}_1 \hat{U}_1^\top - U_1^* U_1^{*\top} \right)$. Then $\hat{A} = A^* + C$ and $rank(C) \leq rank(\hat{U}_1 \hat{U}_1^\top) + rank(U_1^* U_1^{*\top}) \leq 2k$

$\square$

## B.3  PROOF OF THEOREM 3

*Proof.* Since $\mathcal{L}(\hat{A}, \hat{U}) = 0$, we have that for all $t, s \in [T]$,

$$\hat{U}_t \hat{U}_t^T - U_t^* U_t^{*T} = \hat{U}_s \hat{U}_s^T - U_s^* U_s^{*T} \tag{17}$$

Applying this to the first three tasks and rearranging gives that

$$U_1^* U_1^{*T} = \hat{U}_1 \hat{U}_1^T + U_2^* U_2^{*T} - \hat{U}_2 \hat{U}_2^T \tag{18}$$

$$= \hat{U}_1 \hat{U}_1^T + U_3^* U_3^{*T} - \hat{U}_3 \hat{U}_3^T. \tag{19}$$

We first show that $im(\hat{U}_1) = im(U_1^*)$.

Since $U_1^* U_1^{*T} \succeq 0$, we must have that $im(\hat{U}_2) \subseteq im(\hat{U}_1) + im(U_2^*)$ and $im(\hat{U}_3) \subseteq im(\hat{U}_1) + im(U_3^*)$, as otherwise there would exist a vector on $\ker\left( \hat{U}_1 \hat{U}_1^T + U_2^* U_2^{*T} \right) \cap \ker(\hat{U}_2 \hat{U}_2^T)^\perp$ whose existence contradicts the positive semi-definiteness of $U_1^* U_1^{*T}$.

Thus,

$$im(U_1^*) \subseteq im(\hat{U}_1) + im(U_2^*) \tag{20}$$

$$im(U_1^*) \subseteq im(\hat{U}_1) + im(U_3^*) \tag{21}$$

Using that fact that for subspaces $X, Y, Z, X \subseteq Y \implies X + Z \subseteq Y + Z$, we can add $im(U_2^*)$ and $im(U_3^*)$ to both sides of 20 and 21 respectively. This gives that

$$im(U_1^*) \oplus im(U_2^*) \subseteq im(\hat{U}_1) + im(U_2^*) \tag{22}$$

$$im(U_1^*) \oplus im(U_3^*) \subseteq im(\hat{U}_1) + im(U_3^*). \tag{23}$$

For $t \in \{2, 3\}$, we clearly have that $\dim \left( \text{im}(\hat{\boldsymbol{U}}_1) + \text{im}(\boldsymbol{U}_t^*) \right) \leq \dim \text{im}(\hat{\boldsymbol{U}}_1) + \dim \text{im}(\boldsymbol{U}_t^*) \leq 2k$, and $\dim \left( \text{im}(\boldsymbol{U}_1^*) + \text{im}(\boldsymbol{U}_t^*) \right) = 2k$. Thus,

$$(\text{im}(\boldsymbol{U}_1^*) \oplus \text{im}(\boldsymbol{U}_2^*)) = \left( \text{im}(\hat{\boldsymbol{U}}_1) \oplus \text{im}(\boldsymbol{U}_2^*) \right) \tag{24}$$

$$(\text{im}(\boldsymbol{U}_1^*) \oplus \text{im}(\boldsymbol{U}_3^*)) = \left( \text{im}(\hat{\boldsymbol{U}}_1) \oplus \text{im}(\boldsymbol{U}_3^*) \right) \tag{25}$$

**Lemma 2.** $\left( [\text{im}(\hat{\boldsymbol{U}}_1) \oplus \text{im}(\boldsymbol{U}_2^*)] \cap [\text{im}(\hat{\boldsymbol{U}}_1) \oplus \text{im}(\boldsymbol{U}_3^*)] \right) = \text{im}(\hat{\boldsymbol{U}}_1)$

*Proof.* Clearly, $\text{im}(\hat{\boldsymbol{U}}_1) \subseteq \left( [\text{im}(\hat{\boldsymbol{U}}_1) \oplus \text{im}(\boldsymbol{U}_2^*)] \cap [\text{im}(\hat{\boldsymbol{U}}_1) \oplus \text{im}(\boldsymbol{U}_3^*)] \right)$. To show the converse, consider $\boldsymbol{x} \in \left( [\text{im}(\hat{\boldsymbol{U}}_1) \oplus \text{im}(\boldsymbol{U}_2^*)] \cap [\text{im}(\hat{\boldsymbol{U}}_1) \oplus \text{im}(\boldsymbol{U}_3^*)] \right)$.

By assumption there exists some $\boldsymbol{a}, \boldsymbol{b}, \boldsymbol{c}, \boldsymbol{d} \in \mathbb{R}^k$ such that

$$\boldsymbol{x} = \hat{\boldsymbol{U}}_1 \boldsymbol{a} + \boldsymbol{U}_2^* \boldsymbol{b} \tag{26}$$

$$= \hat{\boldsymbol{U}}_1 \boldsymbol{c} + \boldsymbol{U}_3^* \boldsymbol{d} \tag{27}$$

Thus,

$$\hat{\boldsymbol{U}}_1(\boldsymbol{a} - \boldsymbol{c}) + \boldsymbol{U}_2^* \boldsymbol{b} - \boldsymbol{U}_3^* \boldsymbol{d} = \boldsymbol{0}. \tag{28}$$

By Equation 24, we can write

$$\text{im}(\boldsymbol{U}_2^*) = ([\text{im}(\boldsymbol{U}_1^*) \oplus \text{im}(\boldsymbol{U}_2^*)] \cap [\text{im}(\boldsymbol{U}_2^*) \oplus \text{im}(\boldsymbol{U}_3^*)])$$

$$= \left( [\text{im}(\hat{\boldsymbol{U}}_1) \oplus \text{im}(\boldsymbol{U}_2^*)] \cap [\text{im}(\boldsymbol{U}_2^*) \oplus \text{im}(\boldsymbol{U}_3^*)] \right)$$

Thus, $\text{im}(\hat{\boldsymbol{U}}_1) \cap [\text{im}(\boldsymbol{U}_2^*) \oplus \text{im}(\boldsymbol{U}_3^*)] \subseteq \text{im}(\hat{\boldsymbol{U}}_1) \cap \text{im}(\boldsymbol{U}_2^*) = \{\boldsymbol{0}\}$, so

$$\text{im}(\hat{\boldsymbol{U}}_1) \cap [\text{im}(\boldsymbol{U}_2^*) \oplus \text{im}(\boldsymbol{U}_3^*)] = \{\boldsymbol{0}\} \tag{29}$$

Applying Equation (29) to Equation (28) implies that $\boldsymbol{a} = \boldsymbol{c}$ and $\boldsymbol{b} = \boldsymbol{d} = \boldsymbol{0}$. Thus $\boldsymbol{x} = \hat{\boldsymbol{U}}_1 \boldsymbol{a} \in \text{im}(\hat{\boldsymbol{U}}_1)$, so $\left( [\text{im}(\hat{\boldsymbol{U}}_1) \oplus \text{im}(\boldsymbol{U}_2^*)] \cap [\text{im}(\hat{\boldsymbol{U}}_1) \oplus \text{im}(\boldsymbol{U}_3^*)] \right) \subseteq \text{im}(\hat{\boldsymbol{U}}_1)$. $\qquad\square$

Then Equations (20) and (21) combined with Lemma (2) implies that $\text{im}(\boldsymbol{U}_1^*) \subseteq \text{im}(\hat{\boldsymbol{U}}_1)$ but $\dim(\text{im}(\boldsymbol{U}_1^*)) = \dim(\text{im}(\hat{\boldsymbol{U}}_1)) = k$, so $\text{im}(\boldsymbol{U}_1^*) = \text{im}(\hat{\boldsymbol{U}}_1)$.

Since the initial assumptions about $\hat{\boldsymbol{U}}_1$ and $\boldsymbol{U}_1^*$ analogously hold for the corresponding matrices for tasks 2 and 3, by the exact same argument we can show that

$$\text{im}(\boldsymbol{U}_t^*) = \text{im}(\hat{\boldsymbol{U}}_t) \quad \forall t \in [T]. \tag{30}$$

Then by equation (17), $\text{im}(\boldsymbol{U}_1^*) \supseteq \text{im}\left( \hat{\boldsymbol{U}}_1 \hat{\boldsymbol{U}}_1^T - \boldsymbol{U}_1^* \boldsymbol{U}_1^{*T} \right) = \text{im}\left( \hat{\boldsymbol{U}}_2 \hat{\boldsymbol{U}}_2^T - \boldsymbol{U}_2^* \boldsymbol{U}_2^{*T} \right) \subseteq \text{im}(\boldsymbol{U}_2^*)$. Thus,

$$\text{im}\left( \hat{\boldsymbol{U}}_1 \hat{\boldsymbol{U}}_1^T - \boldsymbol{U}_1^* \boldsymbol{U}_1^{*T} \right) \subseteq \text{im}(\boldsymbol{U}_1^*) \cap \text{im}(\boldsymbol{U}_2^*)$$

$$= \{\boldsymbol{0}\}.$$

Thus $\hat{\boldsymbol{U}}_1 \hat{\boldsymbol{U}}_1^T = \boldsymbol{U}_1^* \boldsymbol{U}_1^{*T}$. Then by Equation (17), $\hat{\boldsymbol{U}}_t \hat{\boldsymbol{U}}_t^T = \boldsymbol{U}_t^* \boldsymbol{U}_t^{*T}$ for all $t \in [T]$. Lastly, since $\mathcal{L}(\hat{\boldsymbol{A}}, \hat{\boldsymbol{U}}) = 0$, we have that $\nabla_{\boldsymbol{A}} \mathcal{L}(\hat{\boldsymbol{A}}, \hat{\boldsymbol{U}}) = 0$, so

$$\hat{\boldsymbol{A}} = \boldsymbol{A}^* + \frac{1}{T} \sum_{t=1}^{T} \boldsymbol{U}_t^* \boldsymbol{U}_t^{*\top} - \boldsymbol{U}_t \boldsymbol{U}_t^\top = \boldsymbol{A}^*$$

$\qquad\square$

## B.4 PROOF OF THEOREM 4

Clearly if $\mathcal{L}(\hat{\boldsymbol{A}}, \hat{\boldsymbol{U}}) = 0$, then $(\hat{\boldsymbol{A}}, \hat{\boldsymbol{U}})$ is an SOSP. The reverse direction is the challenging part of the proof. We equivalently prove that if $(\hat{\boldsymbol{A}}, \hat{\boldsymbol{U}})$ is a critical point and $\mathcal{L}(\hat{\boldsymbol{A}}, \hat{\boldsymbol{U}}) \neq 0$, then $\nabla^2 \mathcal{L}(\hat{\boldsymbol{A}}, \hat{\boldsymbol{U}})$ has a negative eigenvalue.

Assume for the sake of contradiction that $(\hat{\boldsymbol{A}}, \hat{\boldsymbol{U}})$ is a critical point and $\mathcal{L}(\hat{\boldsymbol{A}}, \hat{\boldsymbol{U}}) \neq 0$. Then,

$$\nabla_{\boldsymbol{A}} \mathcal{L}(\hat{\boldsymbol{A}}, \hat{\boldsymbol{U}}) = T(\hat{\boldsymbol{A}} - \boldsymbol{A}^*) + \sum_{t=1}^{T} \left( \hat{\boldsymbol{U}}_t \hat{\boldsymbol{U}}_t^\top - \boldsymbol{U}_t^* \boldsymbol{U}_t^{*\top} \right) = \mathbf{0} \tag{31}$$

$$\nabla_{\boldsymbol{U}_t} \mathcal{L}(\hat{\boldsymbol{A}}, \hat{\boldsymbol{U}}) = 2 \left( \hat{\boldsymbol{A}} - \boldsymbol{A}^* + \hat{\boldsymbol{U}}_t \hat{\boldsymbol{U}}_t^\top - \boldsymbol{U}_t^* \boldsymbol{U}_t^{*\top} \right) \hat{\boldsymbol{U}}_t = \mathbf{0} \tag{32}$$

Thus,

$$\hat{\boldsymbol{A}} = \boldsymbol{A}^* - \frac{1}{T} \sum_{t=1}^{T} \left( \hat{\boldsymbol{U}}_t \hat{\boldsymbol{U}}_t^\top - \boldsymbol{U}_t^* \boldsymbol{U}_t^{*\top} \right). \tag{33}$$

Define $\boldsymbol{B}_t(\hat{\boldsymbol{U}}) = \hat{\boldsymbol{U}}_t \hat{\boldsymbol{U}}_t^\top - \boldsymbol{U}_t^* \boldsymbol{U}_t^{*\top} - \frac{1}{T} \sum_{s=1}^{T} \left( \hat{\boldsymbol{U}}_s \hat{\boldsymbol{U}}_s^\top - \boldsymbol{U}_s^* \boldsymbol{U}_s^{*\top} \right)$. Despite being a slight abuse of notation, we refer to $\boldsymbol{B}_t(\hat{\boldsymbol{U}})$ as just $\boldsymbol{B}_t$ for the remainder of the proof.

Then (32) equivalently states:

$$\boldsymbol{B}_t \hat{\boldsymbol{U}}_t = 0. \tag{34}$$

Note that by construction, $\sum_{t=1}^{T} \boldsymbol{B}_t = 0$.

Considering $\mathcal{L}$ as a function of the flattened vector $[\text{vec}(\boldsymbol{A}); \text{vec}(\boldsymbol{U}_1); \text{vec}(\boldsymbol{U}_2)]$, and let $\boldsymbol{U}_1 = [\boldsymbol{x}_1 \ \ldots \ \boldsymbol{x}_k]$, $\boldsymbol{U}_2 = [\boldsymbol{y}_1 \ \ldots \ \boldsymbol{y}_k]$, we compute the Hessian

$$\nabla^2 \mathcal{L} = \begin{bmatrix} \nabla_{\boldsymbol{A}}^2 \mathcal{L} & \nabla_{\boldsymbol{U}_1} \nabla_{\boldsymbol{A}} \mathcal{L} & \nabla_{\boldsymbol{U}_2} \nabla_{\boldsymbol{A}} \mathcal{L} \\ (\nabla_{\boldsymbol{U}_1} \nabla_{\boldsymbol{A}} \mathcal{L})^\top & \nabla_{\boldsymbol{U}_1}^2 \mathcal{L} & \mathbf{0} \\ (\nabla_{\boldsymbol{U}_2} \nabla_{\boldsymbol{A}} \mathcal{L})^\top & \mathbf{0} & \nabla_{\boldsymbol{U}_2}^2 \mathcal{L} \end{bmatrix} \tag{35}$$

where

$$\nabla_{\boldsymbol{A}}^2 \mathcal{L} = 2 \boldsymbol{I}_{d^2}$$

$$\nabla_{\boldsymbol{U}_1} \nabla_{\boldsymbol{A}} \mathcal{L} = [(\boldsymbol{x}_1 \oplus \boldsymbol{x}_1) \ \ldots \ (\boldsymbol{x}_k \oplus \boldsymbol{x}_k)] \in \mathbb{R}^{d^2 \times dk}$$

$$\nabla_{\boldsymbol{U}_2} \nabla_{\boldsymbol{A}} \mathcal{L} = [(\boldsymbol{y}_1 \oplus \boldsymbol{y}_1) \ \ldots \ (\boldsymbol{y}_k \oplus \boldsymbol{y}_k)] \in \mathbb{R}^{d^2 \times dk}$$

$$\nabla_{\boldsymbol{U}_1}^2 \mathcal{L} = 2(\boldsymbol{A} + \boldsymbol{U}_1 \boldsymbol{U}_1^\top - \boldsymbol{A}^* - \boldsymbol{U}_1^* \boldsymbol{U}_1^{*\top}) \otimes \boldsymbol{I}_k$$

$$+ 2 \begin{bmatrix} \boldsymbol{x}_1 \boldsymbol{x}_1^\top + \|\boldsymbol{x}_1\|_2^2 \boldsymbol{I} & \boldsymbol{x}_1^\top \boldsymbol{x}_2 \boldsymbol{I} + \boldsymbol{x}_2 \boldsymbol{x}_1^\top & \ldots & \boldsymbol{x}_1^\top \boldsymbol{x}_k \boldsymbol{I} + \boldsymbol{x}_k \boldsymbol{x}_1^\top \\ \boldsymbol{x}_2^\top \boldsymbol{x}_1 \boldsymbol{I} + \boldsymbol{x}_1 \boldsymbol{x}_2^\top & \boldsymbol{x}_2 \boldsymbol{x}_2^\top + \|\boldsymbol{x}_2\|_2^2 \boldsymbol{I} & \ldots & \boldsymbol{x}_2^\top \boldsymbol{x}_k \boldsymbol{I} + \boldsymbol{x}_k \boldsymbol{x}_2^\top \\ \vdots & \vdots & \ddots & \vdots \\ \boldsymbol{x}_k^\top \boldsymbol{x}_1 \boldsymbol{I} + \boldsymbol{x}_1 \boldsymbol{x}_k^\top & \ldots & \ldots & \boldsymbol{x}_k \boldsymbol{x}_k^\top + \|\boldsymbol{x}_k\|_2^2 \boldsymbol{I} \end{bmatrix}$$

$$\nabla_{\boldsymbol{U}_2}^2 \mathcal{L} = 2(\boldsymbol{A} + \boldsymbol{U}_2 \boldsymbol{U}_2^\top - \boldsymbol{A}^* - \boldsymbol{U}_2^* \boldsymbol{U}_2^{*\top}) \otimes \boldsymbol{I}_k$$

$$+ 2 \begin{bmatrix} \boldsymbol{y}_1 \boldsymbol{y}_1^\top + \|\boldsymbol{y}_1\|_2^2 \boldsymbol{I} & \boldsymbol{y}_1^\top \boldsymbol{y}_2 \boldsymbol{I} + \boldsymbol{y}_2 \boldsymbol{y}_1^\top & \ldots & \boldsymbol{y}_1^\top \boldsymbol{y}_k \boldsymbol{I} + \boldsymbol{y}_k \boldsymbol{y}_1^\top \\ \boldsymbol{y}_2^\top \boldsymbol{y}_1 \boldsymbol{I} + \boldsymbol{y}_1 \boldsymbol{y}_2^\top & \boldsymbol{y}_2 \boldsymbol{y}_2^\top + \|\boldsymbol{y}_2\|_2^2 \boldsymbol{I} & \ldots & \boldsymbol{y}_2^\top \boldsymbol{y}_k \boldsymbol{I} + \boldsymbol{y}_k \boldsymbol{y}_2^\top \\ \vdots & \vdots & \ddots & \vdots \\ \boldsymbol{y}_k^\top \boldsymbol{y}_1 \boldsymbol{I} + \boldsymbol{y}_1 \boldsymbol{y}_k^\top & \ldots & \ldots & \boldsymbol{y}_k \boldsymbol{y}_k^\top + \|\boldsymbol{y}_k\|_2^2 \boldsymbol{I} \end{bmatrix}$$

Note that $\oplus$ denotes the Kronecker sum defined as $\boldsymbol{X} \oplus \boldsymbol{Y} = \boldsymbol{I} \otimes \boldsymbol{X} + \boldsymbol{Y} \otimes \boldsymbol{I}$ where $\otimes$ is the Kronecker product.

**Lemma 3.** $\mathcal{L}(\hat{\boldsymbol{A}}, \hat{\boldsymbol{U}}) = 0$ *if and only if* $\boldsymbol{B}_t = \boldsymbol{0}$ *for each* $t \in [T]$.

*Proof.* Since $(\hat{\boldsymbol{A}}, \hat{\boldsymbol{U}})$ is a critical point, then plugging Equation (33) into the definition of $\mathcal{L}$ gives that

$$\mathcal{L}(\hat{\boldsymbol{A}}, \hat{\boldsymbol{U}}) = \frac{1}{2} \sum_{t=1}^{T} \|\boldsymbol{B}_t\|_F^2 .$$

Thus $\mathcal{L}(\hat{\boldsymbol{A}}, \hat{\boldsymbol{U}}) = 0$ if and only if $\boldsymbol{B}_t = \boldsymbol{0} \quad \forall t$. $\qquad \square$

**Lemma 4.** *If* $\nabla_U^2 \mathcal{L}(\hat{\boldsymbol{A}}, \hat{\boldsymbol{U}}) \succcurlyeq \boldsymbol{0}$, *then the eigenvectors corresponding to the non-zero eigenvalues of* $\hat{\boldsymbol{U}}_t \hat{\boldsymbol{U}}_t^{\top}$ *are the leading non-negative eigenvectors of* $\boldsymbol{A}^* + \boldsymbol{U}_t^* \boldsymbol{U}_t^{*\top} - \hat{\boldsymbol{A}}$ *for all* $t \in [T]$.

*Proof.* Consider the function $\bar{f}_t(\boldsymbol{U}_t; \hat{\boldsymbol{A}}) = \frac{1}{2} \left\| \boldsymbol{A}^* + \boldsymbol{U}_t^* \boldsymbol{U}_t^{*\top} - \hat{\boldsymbol{A}} - \boldsymbol{U}_t \boldsymbol{U}_t^{\top} \right\|_F^2$. $\bar{f}_t$ is simply the $tth$ summand in $\mathcal{L}$ where $\boldsymbol{A} = \hat{\boldsymbol{A}}$ is fixed and we only consider the variable $\boldsymbol{U}_t$. Minimising $\bar{f}_t$ is identical to the problem of symmetric matrix factorization.

Using well-known properties of symmetric matrix factorization, since $\nabla \bar{f}_t(\hat{\boldsymbol{U}}_t) = \boldsymbol{0}$, we must have that $\hat{\boldsymbol{U}}_t = \boldsymbol{V}_t \boldsymbol{\Gamma}$ where the columns of $\boldsymbol{V}_t$ are the properly scaled eigenvectors of $\boldsymbol{A}^* + \boldsymbol{U}_t^* \boldsymbol{U}_t^{*\top} - \hat{\boldsymbol{A}}$ with non-negative eigenvalues where each column has norm equal to the square root of its corresponding eigenvalue, and $\boldsymbol{\Gamma} \in O_k$ is some orthogonal matrix. Further, if the eigenvectors corresponding to the non-zero eigenvalues of $\hat{\boldsymbol{U}}_t \hat{\boldsymbol{U}}_t^{\top}$ are not the leading non-negative eigenvectors, then $\nabla^2 \bar{f}_t(\hat{\boldsymbol{U}}) \not\succcurlyeq \boldsymbol{0}$ by (Zhang et al., 2020). Since $\nabla^2 \bar{f}_t(\hat{\boldsymbol{U}}_t)$ is a diagonal block of $\nabla^2 \mathcal{L}(\hat{\boldsymbol{A}}, \hat{\boldsymbol{U}})$, $\nabla^2 \bar{f}_i(\hat{\boldsymbol{U}}_t) \not\succcurlyeq \boldsymbol{0}$ would imply $\nabla^2 \mathcal{L}(\hat{\boldsymbol{A}}, \hat{\boldsymbol{U}}) \not\succcurlyeq \boldsymbol{0}$. $\qquad \square$

**Remark 2.** *Without loss of generality, we can assume that the eigenvectors corresponding to the non-zero eigenvalues of* $\hat{\boldsymbol{U}}_t \hat{\boldsymbol{U}}_t^{\top}$ *are the leading non-negative eigenvectors of* $\boldsymbol{A}^* + \boldsymbol{U}_t^* \boldsymbol{U}_t^{*\top} - \hat{\boldsymbol{A}}$ *for all* $i$.

**Lemma 5.** $\left( \hat{\boldsymbol{U}}_2 \hat{\boldsymbol{U}}_2^{\top} - \hat{\boldsymbol{U}}_1 \hat{\boldsymbol{U}}_1^{\top} \right) \boldsymbol{x} = \left( \boldsymbol{U}_2^* \boldsymbol{U}_2^{*\top} - \boldsymbol{U}_1^* \boldsymbol{U}_1^{*\top} \right) \boldsymbol{x}$ *for all* $\boldsymbol{x} \in \text{im}(\hat{\boldsymbol{U}}_1) + \text{im}(\hat{\boldsymbol{U}}_2)$.

*Proof.* Recall $\boldsymbol{B}_1 = \frac{1}{2} \left( \hat{\boldsymbol{U}}_1 \hat{\boldsymbol{U}}_1^{\top} - \boldsymbol{U}_1^* \boldsymbol{U}_1^{*\top} - \hat{\boldsymbol{U}}_2 \hat{\boldsymbol{U}}_2^{\top} + \boldsymbol{U}_2^* \boldsymbol{U}_2^{*\top} \right)$. Then applying first-order stationarity and the fact that $\boldsymbol{B}_2 = -\boldsymbol{B}_1$, we have

$$\left( \hat{\boldsymbol{U}}_2 \hat{\boldsymbol{U}}_2^{\top} - \hat{\boldsymbol{U}}_1 \hat{\boldsymbol{U}}_1^{\top} \right) \hat{\boldsymbol{U}}_1 = \left( \boldsymbol{U}_2^* \boldsymbol{U}_2^{*\top} - \boldsymbol{U}_1^* \boldsymbol{U}_1^{*\top} \right) \hat{\boldsymbol{U}}_1$$

$$\left( \hat{\boldsymbol{U}}_2 \hat{\boldsymbol{U}}_2^{\top} - \hat{\boldsymbol{U}}_1 \hat{\boldsymbol{U}}_1^{\top} \right) \hat{\boldsymbol{U}}_2 = \left( \boldsymbol{U}_2^* \boldsymbol{U}_2^{*\top} - \boldsymbol{U}_1^* \boldsymbol{U}_1^{*\top} \right) \hat{\boldsymbol{U}}_2.$$

$\qquad \square$

**Corollary 5.** $\hat{\boldsymbol{U}}_2 \hat{\boldsymbol{U}}_2^{\top} - \hat{\boldsymbol{U}}_1 \hat{\boldsymbol{U}}_1^{\top}$ *and* $\boldsymbol{U}_2^* \boldsymbol{U}_2^{*\top} - \boldsymbol{U}_1^* \boldsymbol{U}_1^{*\top}$ *share an eigenbasis.*

*Proof.* Using the lemma, any non-zero eigenvector-eigenvalue pair of $\hat{\boldsymbol{U}}_2 \hat{\boldsymbol{U}}_2^{\top} - \hat{\boldsymbol{U}}_1 \hat{\boldsymbol{U}}_1^{\top}$ is also an eigenvector-eigenvalue pair of $\boldsymbol{U}_2^* \boldsymbol{U}_2^{*\top} - \boldsymbol{U}_1^* \boldsymbol{U}_1^{*\top}$. Denote the space defined by the span of these eigenvectors as $\boldsymbol{S}$. Then all other eigenvectors of $\boldsymbol{U}_2^* \boldsymbol{U}_2^{*\top} - \boldsymbol{U}_1^* \boldsymbol{U}_1^{*\top}$ are orthogonal to $\boldsymbol{S}$, so they are also 0-eigenvectors of $\hat{\boldsymbol{U}}_2 \hat{\boldsymbol{U}}_2^{\top} - \hat{\boldsymbol{U}}_1 \hat{\boldsymbol{U}}_1^{\top}$. Thus the two matrices share an eigenbasis. $\qquad \square$

**Corollary 6.** $\dim \left( \text{im} \, \hat{\boldsymbol{U}}_1 + \text{im} \, \hat{\boldsymbol{U}}_2 \right) \leq 2k - 1$, *i.e., the set of columns of* $\hat{\boldsymbol{U}}_1$ *and* $\hat{\boldsymbol{U}}_2$ *are not linearly independent.*

*Proof.* Assume for contradiction that the vectors in the set $\boldsymbol{S} = \{\hat{\boldsymbol{U}}_1 \boldsymbol{e}_i \mid i = 1, \ldots, k\} \cup \{\hat{\boldsymbol{U}}_2 \boldsymbol{e}_i \mid i = 1, \ldots, k\}$ are linearly independent, where $\boldsymbol{e}_i$ is the $i$th standard basis vector in $\mathbb{R}^k$.

Then note that $\left( \hat{\boldsymbol{U}}_1 \hat{\boldsymbol{U}}_1^{\top} - \hat{\boldsymbol{U}}_2 \hat{\boldsymbol{U}}_2^{\top} \right) \boldsymbol{x} \neq \boldsymbol{0}$ and $\left( \boldsymbol{U}_1^* \boldsymbol{U}_1^{*\top} - \boldsymbol{U}_2^* \boldsymbol{U}_2^{*\top} \right) \boldsymbol{x} \neq \boldsymbol{0}$ for all $\boldsymbol{x} \in \boldsymbol{S}$. By Lemma (5), $\hat{\boldsymbol{U}}_1 \hat{\boldsymbol{U}}_1^{\top} - \hat{\boldsymbol{U}}_2 \hat{\boldsymbol{U}}_2^{\top}$ and $\boldsymbol{U}_1^* \boldsymbol{U}_1^{*\top} - \boldsymbol{U}_2^* \boldsymbol{U}_2^{*\top}$ agree for each vector on the $2k$-dimensional

space $\mathrm{span}(\boldsymbol{S})$. But, both $\mathrm{rank}(\hat{\boldsymbol{U}}_1\hat{\boldsymbol{U}}_1^\top - \hat{\boldsymbol{U}}_2\hat{\boldsymbol{U}}_2^\top), \mathrm{rank}(\boldsymbol{U}_1^*\boldsymbol{U}_1^{*\top} - \boldsymbol{U}_2^*\boldsymbol{U}_2^{*\top}) \leq 2k$ by construction. Then by dimension counting, $\hat{\boldsymbol{U}}_1\hat{\boldsymbol{U}}_1^\top - \hat{\boldsymbol{U}}_2\hat{\boldsymbol{U}}_2^\top$ and $\boldsymbol{U}_1^*\boldsymbol{U}_1^{*\top} - \boldsymbol{U}_2^*\boldsymbol{U}_2^{*\top}$ must send $\mathrm{span}\{\boldsymbol{S}\}^\perp$ to $\boldsymbol{0}$. Thus, $\hat{\boldsymbol{U}}_1\hat{\boldsymbol{U}}_1^\top - \hat{\boldsymbol{U}}_2\hat{\boldsymbol{U}}_2^\top$ and $\boldsymbol{U}_1^*\boldsymbol{U}_1^{*\top} - \boldsymbol{U}_2^*\boldsymbol{U}_2^{*\top}$ agree on the entire basis formed by concatenating basis vectors of $\mathrm{span}\{\boldsymbol{S}\}^\perp$ with those of $\mathrm{span}(\boldsymbol{S})$. This implies that $\hat{\boldsymbol{U}}_1\hat{\boldsymbol{U}}_1^\top - \hat{\boldsymbol{U}}_2\hat{\boldsymbol{U}}_2^\top = \boldsymbol{U}_1^*\boldsymbol{U}_1^{*\top} - \boldsymbol{U}_2^*\boldsymbol{U}_2^{*\top}$ and thus $\boldsymbol{B}_1 = \hat{\boldsymbol{U}}_1\hat{\boldsymbol{U}}_1^\top - \hat{\boldsymbol{U}}_2\hat{\boldsymbol{U}}_2^\top - \boldsymbol{U}_1^*\boldsymbol{U}_1^{*\top} + \boldsymbol{U}_2^*\boldsymbol{U}_2^{*\top} = \boldsymbol{0}$. Then $\boldsymbol{B}_2 = -\boldsymbol{B}_1 = \boldsymbol{0}$ so by Lemma 3, $\mathcal{L}(\hat{\boldsymbol{A}}, \hat{\boldsymbol{U}}) = 0$ which is a contradiction. $\qquad\square$

**Lemma 6.** $\boldsymbol{U}_2^*\boldsymbol{U}_2^{*\top} - \boldsymbol{U}_1^*\boldsymbol{U}_1^{*\top}$ *has exactly $k$ positive and $k$ negative eigenvalues.*

*Proof.* First, note that $\boldsymbol{U}_2^*\boldsymbol{U}_2^{*\top}$ has exactly $k$ positive eigenvalues and $k - d$ eigenvalues of $\boldsymbol{0}$. Then $\boldsymbol{U}_2^*\boldsymbol{U}_2^{*\top} - (\boldsymbol{U}_1^*\boldsymbol{e}_1)(\boldsymbol{U}_1^*\boldsymbol{e}_1)^\top$ has rank $k + 1$ because of the linear independence of the columns of the combined set of columns $\boldsymbol{U}_1^*$ and $\boldsymbol{U}_2^*$. Further, since we subtract $(\boldsymbol{U}_1^*\boldsymbol{e}_1)(\boldsymbol{U}_1^*\boldsymbol{e}_1)^\top$, we must be accumulating an additional negative eigenvalue relative to $\boldsymbol{U}_2^*\boldsymbol{U}_2^{*\top}$. Continuing this process shows that subtracting $(\boldsymbol{U}_1^*\boldsymbol{e}_{j+1})(\boldsymbol{U}_1^*\boldsymbol{e}_{j+1})^\top$ from $\boldsymbol{U}_2^*\boldsymbol{U}_2^{*\top} - \sum_{t=1}^j (\boldsymbol{U}_1^*\boldsymbol{e}_i)(\boldsymbol{U}_1^*\boldsymbol{e}_i)^\top$ contributes exactly one more negative eigenvalue, since $\boldsymbol{U}_1^*\boldsymbol{e}_{j+1}$ can never be written as a linear combination of $\{\boldsymbol{U}_1^*\boldsymbol{e}_1, \ldots \boldsymbol{U}_1^*\boldsymbol{e}_k, \boldsymbol{U}_2^*\boldsymbol{e}_1, \ldots \boldsymbol{U}_2^*\boldsymbol{e}_j\}$ for $0 < j < k$. The result then follows from induction. $\quad\square$

**Lemma 7.** $\mathrm{rank}(\hat{\boldsymbol{U}}_1) = \mathrm{rank}(\hat{\boldsymbol{U}}_2) = k$.

*Proof.* Assume for contradiction that $\mathrm{rank}(\hat{\boldsymbol{U}}_1) = m < k$ without loss of generality. Since by Remark (2) we assume the columns of $\hat{\boldsymbol{U}}_1$ are the leading $k$ non-negative eigenvectors of $\boldsymbol{A}^* + \boldsymbol{U}_1^*\boldsymbol{U}_1^{*\top} - \hat{\boldsymbol{A}} = \hat{\boldsymbol{U}}_1\hat{\boldsymbol{U}}_1^\top - \boldsymbol{B}_1$, this must imply that $\boldsymbol{A}^* + \boldsymbol{U}_1^*\boldsymbol{U}_1^{*\top} - \hat{\boldsymbol{A}} - \hat{\boldsymbol{U}}_1\hat{\boldsymbol{U}}_1^\top = -\boldsymbol{B}_1 \preccurlyeq \boldsymbol{0}$.

Plugging in the definition of $\boldsymbol{B}_1$ gives that $\frac{1}{2}\left(\hat{\boldsymbol{U}}_1\hat{\boldsymbol{U}}_1^\top - \boldsymbol{U}_1^*\boldsymbol{U}_1^{*\top} - \hat{\boldsymbol{U}}_2\hat{\boldsymbol{U}}_2^\top + \boldsymbol{U}_2^*\boldsymbol{U}_2^{*\top}\right) \succcurlyeq \boldsymbol{0}$. Thus, $\hat{\boldsymbol{U}}_1\hat{\boldsymbol{U}}_1^\top \succcurlyeq \boldsymbol{U}_1^*\boldsymbol{U}_1^{*\top} + \hat{\boldsymbol{U}}_2\hat{\boldsymbol{U}}_2^\top - \boldsymbol{U}_2^*\boldsymbol{U}_2^{*\top} \succcurlyeq \boldsymbol{U}_1^*\boldsymbol{U}_1^{*\top} - \boldsymbol{U}_2^*\boldsymbol{U}_2^{*\top}$. This contradicts the fact from Lemma (6) that $\boldsymbol{U}_1^*\boldsymbol{U}_1^{*\top} - \boldsymbol{U}_2^*\boldsymbol{U}_2^{*\top}$ has $k$ positive eigenvalues. $\qquad\square$

With this lemma, we will prove the existence of a direction of $\nabla^2\mathcal{L}$ with negative curvature. Instead of directly working with this matrix, we instead use the Schur complement to work with a different form.

**Theorem 5.** *(Schur Complement) Since $\nabla_{\boldsymbol{A}}^2\mathcal{L}(\hat{\boldsymbol{A}}, \hat{\boldsymbol{U}}) = 2\boldsymbol{I} \succ \boldsymbol{0}$, $\nabla^2\mathcal{L}(\hat{\boldsymbol{A}}, \hat{\boldsymbol{U}}) \succcurlyeq \boldsymbol{0}$ if and only if*

$$\nabla_{\boldsymbol{U}}^2\mathcal{L}(\hat{\boldsymbol{A}}, \hat{\boldsymbol{U}}) - \left(\nabla_{\boldsymbol{A}}\nabla_{\boldsymbol{U}}\mathcal{L}(\hat{\boldsymbol{A}}, \hat{\boldsymbol{U}})\right)\left(\nabla_{\boldsymbol{A}}^2\mathcal{L}(\hat{\boldsymbol{A}}, \hat{\boldsymbol{U}})\right)^{-1}\left(\nabla_{\boldsymbol{U}}\nabla_{\boldsymbol{A}}\mathcal{L}(\hat{\boldsymbol{A}}, \hat{\boldsymbol{U}})\right) \succcurlyeq \boldsymbol{0}.$$

Define $\boldsymbol{Q} = \nabla_{\boldsymbol{U}}^2\mathcal{L}(\hat{\boldsymbol{A}}, \hat{\boldsymbol{U}}) - \left(\nabla_{\boldsymbol{A}}\nabla_{\boldsymbol{U}}\mathcal{L}(\hat{\boldsymbol{A}}, \hat{\boldsymbol{U}})\right)\left(\nabla_{\boldsymbol{A}}^2\mathcal{L}(\hat{\boldsymbol{A}}, \hat{\boldsymbol{U}})\right)^{-1}\left(\nabla_{\boldsymbol{U}}\nabla_{\boldsymbol{A}}\mathcal{L}(\hat{\boldsymbol{A}}, \hat{\boldsymbol{U}})\right)$.

For example, when $k = 2$ and letting $\boldsymbol{U}_1 = [\boldsymbol{x}_1\ \boldsymbol{x}_2]$, $\boldsymbol{U}_2 = [\boldsymbol{y}_1\ \boldsymbol{y}_2]$, we have

$$\boldsymbol{Q} = \begin{bmatrix} \boldsymbol{Q}_{11} & \boldsymbol{Q}_{12} \\ \boldsymbol{Q}_{12}^\top & \boldsymbol{Q}_{22} \end{bmatrix},$$

where

$$\boldsymbol{Q}_{11} = \begin{bmatrix} 2\boldsymbol{B}_1 + \boldsymbol{x}_1\boldsymbol{x}_1^\top + \|\boldsymbol{x}_1\|_2^2 & \boldsymbol{x}_1^\top\boldsymbol{x}_2\boldsymbol{I} + \boldsymbol{x}_2\boldsymbol{x}_1^\top \\ \boldsymbol{x}_2^\top\boldsymbol{x}_1\boldsymbol{I} + \boldsymbol{x}_1\boldsymbol{x}_2^\top & 2\boldsymbol{B}_1 + \boldsymbol{x}_2\boldsymbol{x}_2^\top + \|\boldsymbol{x}_2\|_2^2 \end{bmatrix}$$

$$\boldsymbol{Q}_{12} = \begin{bmatrix} -\boldsymbol{x}_1^\top\boldsymbol{y}_1\boldsymbol{I} - \boldsymbol{y}_1\boldsymbol{x}_1^\top & -\boldsymbol{x}_1^\top\boldsymbol{y}_2\boldsymbol{I} - \boldsymbol{y}_2\boldsymbol{x}_1^\top \\ -\boldsymbol{x}_2^\top\boldsymbol{y}_1\boldsymbol{I} - \boldsymbol{y}_1\boldsymbol{x}_2^\top & \boldsymbol{x}_2^\top\boldsymbol{y}_2\boldsymbol{I} - \boldsymbol{y}_2\boldsymbol{x}_2^\top \end{bmatrix}$$

$$\boldsymbol{Q}_{22} = \begin{bmatrix} 2\boldsymbol{B}_2 + \boldsymbol{y}_1\boldsymbol{y}_1^\top + \|\boldsymbol{y}_1\|_2^2 & \boldsymbol{y}_1^\top\boldsymbol{y}_2\boldsymbol{I} + \boldsymbol{y}_2\boldsymbol{y}_1^\top \\ \boldsymbol{y}_2^\top\boldsymbol{y}_1\boldsymbol{I} + \boldsymbol{y}_1\boldsymbol{y}_2^\top & 2\boldsymbol{B}_2 + \boldsymbol{y}_2\boldsymbol{y}_2^\top + \|\boldsymbol{y}_2\|_2^2 \end{bmatrix}$$

For brevity, we do not include the full form of $\boldsymbol{Q}$ for general $k$. However, we can make an easy simplification that will allow for a much cleaner expression.

Using Corollaries (5) and (6), there is an eigenvector $\boldsymbol{z}$ of $\boldsymbol{U}_2^* \boldsymbol{U}_2^{*\top} - \boldsymbol{U}_1^* \boldsymbol{U}_1^{*\top}$ with eigenvalue $\lambda \neq 0$ such that $\boldsymbol{z} \in \ker\left(\hat{\boldsymbol{U}}_2 \hat{\boldsymbol{U}}_2^\top - \hat{\boldsymbol{U}}_1 \hat{\boldsymbol{U}}_1^\top\right)$. Assume without loss of generality that $\lambda > 0$, and consider $\boldsymbol{\alpha} \in \mathbb{R}^{2k}$. Define the function $g(\cdot\,;\boldsymbol{z}) : \mathbb{R}^{2k} \to \mathbb{R}$ parameterized by $\boldsymbol{z}$ such that $g(\boldsymbol{\alpha};\boldsymbol{z}) = (\boldsymbol{\alpha} \otimes \boldsymbol{z})^\top \boldsymbol{Q}\,(\boldsymbol{\alpha} \otimes \boldsymbol{z})$, where we partition $\boldsymbol{\alpha} = [\boldsymbol{\alpha}_1; \boldsymbol{\alpha}_2]$, $\boldsymbol{\alpha}_1, \boldsymbol{\alpha}_2 \in \mathbb{R}^k$. Then after some algebra,

$$g\left(\boldsymbol{\alpha};\boldsymbol{z}\right) = \left\|\hat{\boldsymbol{U}}_1 \boldsymbol{\alpha}_1 + \hat{\boldsymbol{U}}_2 \boldsymbol{\alpha}_2\right\|_2^2 + \lambda\left(\|\boldsymbol{\alpha}_1\|_2^2 - \|\boldsymbol{\alpha}_2\|_2^2\right). \tag{36}$$

We prove the existence of $\boldsymbol{\alpha} \in \mathbb{R}^{2k}, \boldsymbol{x} \in \mathbb{R}^d$ such that $g\left(\boldsymbol{\alpha};\boldsymbol{x}\right) < 0$ considering two different cases. Define $N^- : S_d \to \mathbb{Z}$ as the function that returns the number of negative eigenvalues of its input.

**Case 1**: $N^-\left(\hat{\boldsymbol{U}}_2 \hat{\boldsymbol{U}}_2^\top - \hat{\boldsymbol{U}}_1 \hat{\boldsymbol{U}}_1^\top\right) < k$.

Using Corollary (6), we can pick $\boldsymbol{\alpha}$ such that $\hat{\boldsymbol{U}}_1 \boldsymbol{\alpha}_1 + \hat{\boldsymbol{U}}_2 \boldsymbol{\alpha}_2 = 0$, $\boldsymbol{\alpha}_1, \boldsymbol{\alpha}_2 \neq 0$.

Because $N^-\left(\hat{\boldsymbol{U}}_2 \hat{\boldsymbol{U}}_2^\top - \hat{\boldsymbol{U}}_1 \hat{\boldsymbol{U}}_1^\top\right) < k$, $N^-\left(\boldsymbol{U}_2^* \boldsymbol{U}_2^{*\top} - \boldsymbol{U}_1^* \boldsymbol{U}_1^{*\top}\right) = k$, and $\hat{\boldsymbol{U}}_2 \hat{\boldsymbol{U}}_2^\top - \hat{\boldsymbol{U}}_1 \hat{\boldsymbol{U}}_1^\top$ and $\boldsymbol{U}_2^* \boldsymbol{U}_2^{*\top} - \boldsymbol{U}_1^* \boldsymbol{U}_1^{*\top}$ share an eigenbasis by Corollary 5, there exists $\boldsymbol{z}^- \in \mathbb{R}^d$ that is a $\lambda^-$-eigenvector of $\boldsymbol{U}_2^* \boldsymbol{U}_2^{*T} - \boldsymbol{U}_1^* \boldsymbol{U}_1^{*T}, \lambda^- < 0$, where $\boldsymbol{z} \in \ker\left(\hat{\boldsymbol{U}}_2 \hat{\boldsymbol{U}}_2^\top - \hat{\boldsymbol{U}}_1 \hat{\boldsymbol{U}}_1^\top\right)$

Then for the same choice of $\boldsymbol{\alpha}$,

$$\operatorname{sign}\left(g\left(\boldsymbol{\alpha};\boldsymbol{z}\right)\right) = \operatorname{sign}\left(\|\boldsymbol{\alpha}_1\|_2^2 - \|\boldsymbol{\alpha}_2\|_2^2\right)$$

$$\operatorname{sign}\left(g\left(\boldsymbol{\alpha};\boldsymbol{z}^-\right)\right) = \operatorname{sign}\left(\|\boldsymbol{\alpha}_2\|_2^2 - \|\boldsymbol{\alpha}_1\|_2^2\right).$$

Then if $\|\boldsymbol{\alpha}_1\|_2 \neq \|\boldsymbol{\alpha}_2\|_2$, one of the above expressions is negative and thus $\boldsymbol{Q}$ has a negative eigenvalue. This then implies $\nabla^2 \mathcal{L}(\hat{\boldsymbol{A}}, \hat{\boldsymbol{U}}) \nsucceq 0$.

Otherwise $\|\boldsymbol{\alpha}_1\|_2 = \|\boldsymbol{\alpha}_2\|_2$. Then $g\left(\boldsymbol{\alpha};\boldsymbol{z}\right) = 0$, but $\nabla_{\boldsymbol{\alpha}_1} g(\boldsymbol{\alpha};\boldsymbol{z}) = \hat{\boldsymbol{U}}_1^\top\left(\hat{\boldsymbol{U}}_1 \bar{\boldsymbol{\alpha}}_1 + \hat{\boldsymbol{U}}_2 \boldsymbol{\alpha}_2\right) - 2\lambda \boldsymbol{\alpha}_2 = -2\lambda \boldsymbol{\alpha}_2 \neq 0$. Thus $g(\boldsymbol{\alpha};\boldsymbol{z}) = 0$ and $\nabla g(\boldsymbol{\alpha};\boldsymbol{z}) \neq 0$ so there exists $\bar{\boldsymbol{\alpha}}$ in an infinitesimal neighborhood around $\boldsymbol{\alpha}$ where $g(\bar{\boldsymbol{\alpha}};\boldsymbol{z}) < 0$. Thus $\boldsymbol{Q}$ has a negative eigenvalue so $\nabla^2 \mathcal{L}(\hat{\boldsymbol{A}}, \hat{\boldsymbol{U}}) \nsucceq 0$.

**Case 2**: $N^-\left(\hat{\boldsymbol{U}}_2 \hat{\boldsymbol{U}}_2^\top - \hat{\boldsymbol{U}}_1 \hat{\boldsymbol{U}}_1^\top\right) = k$.

Define $m = \dim\left(\operatorname{im}(\hat{\boldsymbol{U}}_1) \cap \operatorname{im}(\hat{\boldsymbol{U}}_2)\right)$. By Corollary 6, $m \geq 1$, so we can select orthogonal matrix $\boldsymbol{\Gamma} \in O_k$ such that $\hat{\boldsymbol{U}}_2 \boldsymbol{\Gamma} \boldsymbol{e}_1 \in \left(\operatorname{im}(\hat{\boldsymbol{U}}_1) \cap \operatorname{im}(\hat{\boldsymbol{U}}_2)\right)$. Define $\boldsymbol{y} = \hat{\boldsymbol{U}}_2 \boldsymbol{\Gamma} \boldsymbol{e}_1$.

Clearly for any $\boldsymbol{B} \in S_d$ and $\boldsymbol{R} \in S_d^+$, $N^-(\boldsymbol{B}) \geq N^-(\boldsymbol{B} + \boldsymbol{R})$. Then since $N^-\left(-\hat{\boldsymbol{U}}_1 \hat{\boldsymbol{U}}_1^\top\right) = k$ by Lemma (7), we have that

$$k = N^-(-\hat{\boldsymbol{U}}_1 \hat{\boldsymbol{U}}_1^\top) \geq N^-(\boldsymbol{y}\boldsymbol{y}^\top - \hat{\boldsymbol{U}}_1 \hat{\boldsymbol{U}}_1^\top) = N^-\left(\left(\hat{\boldsymbol{U}}_2 \boldsymbol{\Gamma} \boldsymbol{e}_1\right)\left(\hat{\boldsymbol{U}}_2 \boldsymbol{\Gamma} \boldsymbol{e}_1\right)^\top - \hat{\boldsymbol{U}}_1 \hat{\boldsymbol{U}}_1^\top\right)$$

$$\geq N^-\left(\left(\hat{\boldsymbol{U}}_2 \boldsymbol{\Gamma}\right)\left(\hat{\boldsymbol{U}}_2 \boldsymbol{\Gamma}\right)^\top - \hat{\boldsymbol{U}}_1 \hat{\boldsymbol{U}}_1^\top\right) = N^-\left(\hat{\boldsymbol{U}}_2 \hat{\boldsymbol{U}}_2^\top - \hat{\boldsymbol{U}}_1 \hat{\boldsymbol{U}}_1^\top\right) = k,$$

Thus, $N^-(\boldsymbol{y}\boldsymbol{y}^\top - \hat{\boldsymbol{U}}_1 \hat{\boldsymbol{U}}_1^\top) = k$. But, since $\boldsymbol{y} \in \operatorname{im}(\hat{\boldsymbol{U}}_1)$, $\operatorname{rank}\left(\boldsymbol{y}\boldsymbol{y}^\top - \hat{\boldsymbol{U}}_1 \hat{\boldsymbol{U}}_1^\top\right) = k$. Thus,

$$\boldsymbol{y}\boldsymbol{y}^\top - \hat{\boldsymbol{U}}_1 \hat{\boldsymbol{U}}_1^\top \preccurlyeq 0. \tag{37}$$

Take $\boldsymbol{\alpha}$ such that $\hat{\boldsymbol{U}}_1 \boldsymbol{\alpha}_1 = -\boldsymbol{y}$ and $\boldsymbol{\alpha}_2 = \boldsymbol{\Gamma} \boldsymbol{e}_1$. Then

$$\boldsymbol{y}_1 \boldsymbol{y}_1^\top - \hat{\boldsymbol{U}}_1 \hat{\boldsymbol{U}}_1^\top = \left(\hat{\boldsymbol{U}}_1 \boldsymbol{\alpha}\right)\left(\hat{\boldsymbol{U}}_1 \boldsymbol{\alpha}\right)^\top - \hat{\boldsymbol{U}}_1 \hat{\boldsymbol{U}}_1^\top \tag{38}$$

$$= \hat{\boldsymbol{U}}_1\left(\boldsymbol{\alpha}_1 \boldsymbol{\alpha}_1^\top - \boldsymbol{I}\right)\hat{\boldsymbol{U}}_1^\top \preccurlyeq 0. \tag{39}$$

Therefore $\|\boldsymbol{\alpha}_1\|_2 \le 1$.

Then $g(\boldsymbol{\alpha}; \boldsymbol{z}) = \left\|\hat{\boldsymbol{U}}_1\boldsymbol{\alpha}_1 + \hat{\boldsymbol{U}}_2\boldsymbol{\alpha}_2\right\|_2^2 + \lambda\left(\|\boldsymbol{\alpha}_1\|_2^2 - \|\boldsymbol{\alpha}_2\|_2^2\right) = \lambda\left(\|\boldsymbol{\alpha}_1\|_2^2 - 1\right) \le 0.$

If $g(\boldsymbol{\alpha}; \boldsymbol{z}) < 0$, then we are done. Otherwise, $g(\boldsymbol{\alpha}; \boldsymbol{z}) = 0$. Then the same analysis from Case 1 will show that $\nabla g(\boldsymbol{\alpha}; \boldsymbol{z}) \ne \boldsymbol{0}$, so there exists $\bar{\boldsymbol{\alpha}}$ in an infinitesimal neighborhood around $\boldsymbol{\alpha}$ where $g(\bar{\boldsymbol{\alpha}}; \boldsymbol{z})$ is strictly negative. This then implies our desired result.

## B.5 DERIVATION OF EQUATION (6)

Recall our generative model $\boldsymbol{x} \sim \mathcal{N}(\boldsymbol{0}, \sigma_x^2\boldsymbol{I})$, $\boldsymbol{\epsilon} \sim \mathcal{N}(\boldsymbol{0}, \sigma_\epsilon^2\boldsymbol{I})$, and $\boldsymbol{y} = \boldsymbol{A}_t^*\boldsymbol{x} + \boldsymbol{\epsilon}$, where $\boldsymbol{x}$ and $\boldsymbol{\epsilon}$ are independent. Then,

$$
\begin{aligned}
2\mathbb{E}[\mathcal{L}_t^1(\boldsymbol{A}_t)] &= \mathbb{E}\left[\|\boldsymbol{y} - \boldsymbol{A}_t\boldsymbol{x}\|_2^2\right] \\
&= \mathbb{E}\left[\|\boldsymbol{A}_t^*\boldsymbol{x} + \boldsymbol{\epsilon} - \boldsymbol{A}_t\boldsymbol{x}\|_2^2\right] \\
&= \mathbb{E}\left[\|(\boldsymbol{A}_t^* - \boldsymbol{A}_t)\boldsymbol{x} + \boldsymbol{\epsilon}\|_2^2\right] \\
&= \mathbb{E}\left[\left(\|(\boldsymbol{A}_t^* - \boldsymbol{A}_t)\boldsymbol{x}\|_2^2 + \|\boldsymbol{\epsilon}\|_2^2 + 2\boldsymbol{\epsilon}^\top(\boldsymbol{A}_t^* - \boldsymbol{A}_t)\boldsymbol{x}\right)\right] \\
&= \mathbb{E}\left[\boldsymbol{x}^\top(\boldsymbol{A}_t^* - \boldsymbol{A}_t)^\top(\boldsymbol{A}_t^* - \boldsymbol{A}_t)\boldsymbol{x}\right] + \mathbb{E}\left[\|\boldsymbol{\epsilon}\|_2^2\right] + 2\mathbb{E}\left[\boldsymbol{\epsilon}^\top(\boldsymbol{A}_t^* - \boldsymbol{A}_t)\boldsymbol{x}\right] \\
&= \mathbb{E}\left[\operatorname{tr}\left(\boldsymbol{x}^\top(\boldsymbol{A}_t^* - \boldsymbol{A}_t)^\top(\boldsymbol{A}_t^* - \boldsymbol{A}_t)\boldsymbol{x}\right)\right] + \sigma_\epsilon^2 + 2\mathbb{E}\left[\boldsymbol{\epsilon}\right]^\top(\boldsymbol{A}_t^* - \boldsymbol{A}_t)\mathbb{E}\left[\boldsymbol{x}\right] \quad (\boldsymbol{\epsilon}, \boldsymbol{x} \text{ are independent}) \\
&= \mathbb{E}\left[\operatorname{tr}\left((\boldsymbol{A}_t^* - \boldsymbol{A}_t)^\top(\boldsymbol{A}_t^* - \boldsymbol{A}_t)\boldsymbol{x}\boldsymbol{x}^\top\right)\right] + \sigma_\epsilon^2 \quad (\text{by cyclic property of trace and since } \mathbb{E}[\boldsymbol{x}] = 0) \\
&= \operatorname{tr}\left((\boldsymbol{A}_t^* - \boldsymbol{A}_t)^\top(\boldsymbol{A}_t^* - \boldsymbol{A}_t)\mathbb{E}\left[\boldsymbol{x}\boldsymbol{x}^\top\right]\right) + \sigma_\epsilon^2 \\
&= \sigma_x^2 \operatorname{tr}\left((\boldsymbol{A}_t^* - \boldsymbol{A}_t)^\top(\boldsymbol{A}_t^* - \boldsymbol{A}_t)\right) + \sigma_\epsilon^2 \\
&= \sigma_x^2 \|\boldsymbol{A}_t^* - \boldsymbol{A}_t\|_F^2 + \sigma_\epsilon^2
\end{aligned}
$$

Thus, $\mathbb{E}[\mathcal{L}_t^1(\boldsymbol{A}_t)] = \frac{1}{2}\left(\sigma_x^2\|\boldsymbol{A}_t^* - \boldsymbol{A}_t\|_F^2 + \sigma_\epsilon^2\right)$. Then $\mathbb{E}[\mathcal{L}_t^N(\boldsymbol{A}_t)] = \mathbb{E}[\mathcal{L}_t^1(\boldsymbol{A}_t)]$ by linearity of expectation, so

$$
\frac{1}{2}\left\|\boldsymbol{A}^* + \boldsymbol{U}_t^*\boldsymbol{U}_t^{*\top} - \boldsymbol{A}_t\right\|_F^2 = \frac{1}{\sigma_x^2}\left(\mathbb{E}\left[\mathcal{L}_t^N(\boldsymbol{A}_t)\right] - \frac{\sigma_\epsilon^2}{2}\right)
$$

## C    LLM Training Hyperparameters

| Hyperparameter | Standard Retraining | Meta-LoRA-8 | Meta-LoRA-16 |
|---|---|---|---|
| Learning Rate | 5e-5 | 3e-5 | 5e-5 |
| Learning Rate Schedule | Linear | Linear | Linear |
| Batch Size | 6 | 4 | 4 |
| Epochs | 30 | 30 | 30 |
| Optimizer | AdamW | AdamW | AdamW |
| LoRA Rank | N/A | 8 | 16 |
| LoRA Dropout | N/A | 0.1 | .1 |
| LoRA Alpha | N/A | 16 | 16 |

Table 2: Retraining Hyperparameters

| Hyperparameter | Rank-$k$ LoRA Fine-Tuning |
|---|---|
| Learning Rate | 3e-5 |
| Learning Rate Schedule | Linear |
| Batch Size | 6 |
| Epochs | 30 |
| Optimizer | AdamW |
| LoRA Rank | k |
| LoRA Dropout | .1 |
| LoRA Alpha | 16 |

Table 3: Rank-$k$ LoRA Fine-Tuning Hyperparameters, $k \in \{8, 16\}$

### C.1    Note on Number of Trainable Parameters

For simplicity assume our model architecture consisted of $m$ layers, where each layer was parameterized by a $d \times d$ matrix, and we use rank-$k$ adaptations for each layer for our Meta-LoRA objective, where $k \ll d$. Then the standard retraining method uses $md^2$ trainable parameters, while minimizing the Meta-LoRA objective uses $m(d^2 + 2kdT)$ trainable parameters. Although Meta-LoRA uses some additional parameters, since $k$ is small relative to $d$ and we work in the setting where $k(T + 1) < d$, asymptotically $m(d^2 + 2kdT) = O(md^2)$ so the increase in trainable parameters is minor. After running either of these retraining procedures, the fine-tuning stages are identical and require the same number of trainable parameters no matter which retraining procedure was run.

## D    Theory Notes

### D.1    Non-Uniqueness of Global Min for $T = 2$

Consider $T = 2$, $k = 1$, $d = 2$, $\boldsymbol{A}^* = \boldsymbol{0}$, and $\boldsymbol{u}_t^* = \boldsymbol{e}_t$ for $t = 1, 2$, where $\boldsymbol{e}_t$ is the $t_{th}$ standard basis vector. Clearly the ground truth perturbations $\boldsymbol{u}_i^*$ are orthonormal and thus linearly independent. The set of global minima of $\mathcal{L}$ are $(\boldsymbol{A}, \boldsymbol{U})$ such that $\boldsymbol{A} = \frac{1}{T} \sum_{t=1}^{T} \left( \boldsymbol{u}_t^* \boldsymbol{u}_t^{*\top} - \boldsymbol{u}_t \boldsymbol{u}_t^\top \right)$ and $\boldsymbol{u}_t \boldsymbol{u}_t^\top - \boldsymbol{u}_t^* \boldsymbol{u}_t^{*\top} - \frac{1}{T} \sum_{s=1}^{T} \left( \boldsymbol{u}_s \boldsymbol{u}_s^\top - \boldsymbol{u}_s^* \boldsymbol{u}_s^{*\top} \right) = \boldsymbol{0}$. It is not hard to see that a global minimum follows from any set values of $\boldsymbol{u}_1, \boldsymbol{u}_2$ such that $\boldsymbol{u}_1 \boldsymbol{u}_1^\top - \boldsymbol{u}_2 \boldsymbol{u}_2^\top = \begin{bmatrix} 1 & 0 \\ 0 & -1 \end{bmatrix}$. When properly parameterized, this system of equations defines a hyperbola where each point corresponds to a global minimum of $\mathcal{L}$.

### D.2    Spurious Local Minima

We observe that for $T \geq 3$, for certain tasks $\boldsymbol{U}^* = (\boldsymbol{U}_1^*, \boldsymbol{U}_2^*, \boldsymbol{U}_3^*)$, it is possible to find points $\boldsymbol{U}$ that are local minima, but not global minima. To find these points, we sample true tasks $\boldsymbol{U}^*$ from a

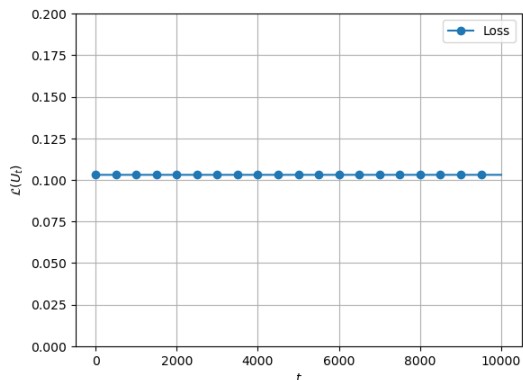

Figure 2: Loss does not decrease near these spurious local minima

normal distribution and use a numerical solver to find zeros of the gradient of the reduced loss

$$\hat{\mathcal{L}}(\boldsymbol{U}) = \sum_{t=1}^{T} \left\| \boldsymbol{U}_t \boldsymbol{U}_t^\top - \boldsymbol{U}_t^* \boldsymbol{U}_t^{*\top} - \frac{1}{T} \sum_{s=1}^{T} (\boldsymbol{U}_s \boldsymbol{U}_s^\top - \boldsymbol{U}_s^* \boldsymbol{U}_s^{*\top}) \right\|_F^2.$$

Through the Schur complement argument used to prove Theorem 4, we can see that $\hat{\mathcal{L}}$ has a spurious local minimum only if $\mathcal{L}$ has a spurious local minimum.

Typically, these zeros are close to the global minimum. Occasionally, it is possible to find a point $\hat{\boldsymbol{U}}$ with gradients close to $0$ and with positive definite Hessians. We then confirm that these are close to the spurious local minimum through the following argument.

Consider the function
$$r(\boldsymbol{U}) = \text{vec}(\boldsymbol{U} - \hat{\boldsymbol{U}})^\top \text{vec}(\nabla \hat{\mathcal{L}}(\boldsymbol{U})).$$

Clearly, there is a minimum of $\hat{\mathcal{L}}$ in the $\delta$-ball of $\hat{\boldsymbol{U}}$ if $r(\boldsymbol{U}) > 0$ for all $\boldsymbol{U}$ on the boundary of the $\delta$-ball. As $r$ is continuous, if for some small enough $\epsilon, \gamma > 0$ if $r(\boldsymbol{U}) > \gamma > 0$ for all $\boldsymbol{U}$ on the $\epsilon$-net of the boundary of the $\delta$-ball, then there exists a spurious local minimum in the $\delta$-ball around $\hat{\boldsymbol{U}}$. Numerically, such points and $\epsilon, \delta$, and $\gamma$ can be found which would imply that spurious local minima exist, barring any errors due to numerical computation. To confirm, we run gradient descent from this point and observe that the loss stays constant.

## E    EXAMPLE PSEUDOCODE FOR MINIMIZING (4)

---
**Algorithm 1** Meta-Adapter Training

---
1: **Input:** Tasks $\mathcal{T}_t$, $t \in [T]$, learning rate $\eta$, number of epochs $N_e$, batches per epoch $N_b$

2: **Initialize:** Model parameters $\boldsymbol{W}_0, \boldsymbol{\theta}_0^{(t)}$ for all $t = 1, \ldots, T$

3: **for** epoch $e = 1$ to $N_e$ **do**

4:     **for** $b = 1, \ldots, N_b$ **do**

5:         **for** $t = 1, \ldots, T$ **do**

6:             Load next batch $\beta_{t,b}$ from $\mathcal{T}_i$

7:             Compute gradient $\boldsymbol{g}^{(t)} = \nabla_{\boldsymbol{W},\boldsymbol{\theta}^{(t)}} \left( \sum_{(\boldsymbol{x},\boldsymbol{y}) \in \beta_{t,b}} \mathcal{L}\left( \left( \Phi_{\text{FT}}\left( \boldsymbol{x} \, ; \boldsymbol{W}, \boldsymbol{\theta}^{(t)} \right), \boldsymbol{y} \right) \right) \right)$

8:             Update adapter parameters: $\boldsymbol{\theta}_{e+1}^{(t)} \leftarrow \boldsymbol{\theta}_e^{(t)} - \eta_e \boldsymbol{g}_{\boldsymbol{\theta}^{(t)}}$

9:         **end for**

10:         Update base parameters: $\boldsymbol{W}_{e+1} \leftarrow \boldsymbol{W}_e - \eta_e \sum_{t=1}^{T} \boldsymbol{g}_{\boldsymbol{W}}^{(t)}$

11:     **end for**

12: **end for**

---

