# OpenReview forum: "Meta-Learning Adaptable Foundation Models"
_ICLR.cc/2025/Conference — Submitted to ICLR 2025_

### Official Review · Reviewer_NxBk · 2024-11-02

**Soundness:** 2
**Presentation:** 3
**Contribution:** 2
**Rating:** 5
**Confidence:** 4

**Summary:**

This paper studies the problem of supervised fine-tuning or retraining of foundation models and proposes an adaptation-aware objective that is trained using the fact that the final downstream purpose of retraining is to be adapted using a PEFT method like LoRA. They show that the standard objective will not require the optimal parameter in linear model setting while their objective will (under assumptions about infinite samples per task). The validate their findings on experiments in both the linear model and Transformer settings.

**Strengths:**

1. The problem studied is highly relevant and brings in ideas from meta-learning towards the analysis of fine-tuning of FMs. It is interesting to study whether an adaptation-aware objective can bring provable benefits.
2. The authors make progress by showing that, under a specific model, a meta-learning-style objective does better at parameter recovery.
3. The authors validate their findings experimentally, and the theoretical/empirical optimization result at T=2 and T=3 is interesting.

**Weaknesses:**

1. The lower-bound for “standard retraining” is stated in terms of rank, which does not seem very convincing. For example, the rank could be high but the norm difference between \hat A_{SR} and A^* could be very small (e.g O(1/T)) in which case we would still be doing fine with standard retraining. For example, Theorem 1 and Corollary 1 do not imply that “The optimal test error even scales with T.”
2. I do not understand how the “infinite sample loss” (Equation 6) is derived. It is defined as the expectation of a finite sample loss L_t^N, but then the limit as N->\infty is not taken. Furthermore, even at infinite samples the underlying noise in the model should still be present after taking expectations and a limit, but it is absent. Somehow the loss can be zero, which is better than the Bayes’ risk.
3. It is somewhat dissatisfying that the proposed objective and analysis largely depends fully on doing some type of low-rank adaptation. At full rank the Meta-LoRA objective is not useful, but in-principle an effective objective will interpolate smoothly between being able to do low-rank and full-rank adaptation effectively. In practice it does not seem like LoRA is useful for purposes of model capacity control, only to reduce fine-tuning memory usage,.
4. The model demonstrates no benefit of being useful beyond 3 tasks, despite this presumably being useful in practice. It seems the main purpose of optimization here is parameter recovery rather than statistical learning.
5. Code is not provided for the experiments.

**Questions:**

1. The abstract describes the FM paradigm as a two-phase procedure but the intro describes it as a three-phase procedure (it seems the abstract does not consider retraining a separate step).
2. Remark 1 alludes to the “generation process of each U_t^*” but I could not find any description of how those matrices are generated. The same remark also states that “k(T+1)\ll d” but I could not find that assumption anywhere (and while perhaps fine for theory, in practice it is not always true).

---

> ### Author Response · Authors · 2024-11-14
>
> **Q: The lower-bound for “standard retraining” is stated in terms of rank, which does not seem very convincing. For example, the rank could be high but the norm difference between $\\hat A\_{SR}$ and $A^\ast$ could be very small (e.g O(1/T)) in which case we would still be doing fine with standard retraining. For example, Theorem 1 and Corollary 1 do not imply that “The optimal test error even scales with T.”**
>
> **A.** Thank you for the comment. We chose to present the error in terms of rank as that requires the fewest assumptions in our statement. Although in general a large difference in rank may not imply a large difference in norm, for our generative model it is the case that this rank difference translates to a substantial difference in norm. According to our generative model, the columns of each $\mathbf{U}i$ are i.i.d. Gaussian random vectors with mean $\mathbf{0}$ and covariance $I_d$.
> Then, $||{\hat A_{SR}-A^*}|| = ||{\frac{1}{T} \sum_{t}^T U_t^\ast {U_t^\ast}^\top}|| = k\sqrt{d}$ almost surely as $T$ goes to infinity, where all matrix norms are Frobenius norms. Further, the test in error for test task adaptation rank $k'$ as defined in Equation 7 scales as $(d-k')k^2$, so standard retraining would still incur a significant test error. We have added a corollary to explain this result and updated our discussion after the statement of Corollary 1 to be more precise. The derivations of these results are added in the Appendix B.1.
>
> **Q: I do not understand how the “infinite sample loss” (Equation 6) is derived. It is defined as the expectation of a finite sample loss $\mathcal{L}_t^N$, but then the limit as $N\to\infty$ is not taken. Furthermore, even at infinite samples the underlying noise in the model should still be present after taking expectations and a limit, but it is absent. Somehow the loss can be zero, which is better than the Bayes’ risk.**
>
> **A.** Thank you for making this point. We first would like to clarify that taking the expectation of $\\mathcal{L}\_t^N$ is equivalent to taking the limit as $N$ goes to infinity and the noise term will reduce to a constant. To address the first part, note that $\\mathcal{L}\_t ^N (A\_t)$ is the $N$-sample empirical average of the random variable $\\frac{1}{2} \\|{y - A_t x}\\|\_{2}^2$, where the randomness is generated from $x \\sim \\mathcal{N}(0,\\sigma_x^2 I)$ and $\epsilon \sim \mathcal{N}(0,\sigma_{\epsilon}^2 I)$, since $y = A_t^* x + \epsilon$. Then by the Strong Law of Large Numbers, $\lim\limits_{N \rightarrow \infty} \mathcal{L}_t^N = \\mathbb{E}[\\mathcal{L}_t^1]$ almost surely. Then by linearity of expectation, $\mathbb{E}[\\mathcal{L}\_t^1] = \\mathbb{E}[\\mathcal{L}\_t^N]$ for any natural number $N$. Thus, taking the infinite sample loss is equivalent to taking an expectation. In this case, the noise will reduce to a constant and the infinite sample loss $\\mathbb{E}[\\mathcal{L}\_t^N(A_t)]$ is equal to $\\|{A_t^* - A_t}\\|\_{F}^2$ up to constant factors. The derivation of this has been added in Appendix B.5. Thank you for the comment that $\\mathbb{E}[\\mathcal{L}\_t^N(A_t)]$ cannot be zero due to the noise floor caused by the variance of \beps. We updated our statement of Equation 6 to be more precise to account for this fact, and we note that for optimization purposes we can ignore this constant additive factor.

---

> > ### Author Response · Authors · 2024-11-14
> >
> > **Q:  It is somewhat dissatisfying that the proposed objective and analysis largely depends fully on doing some type of low-rank adaptation. At full rank the Meta-LoRA objective is not useful, but in-principle an effective objective will interpolate smoothly between being able to do low-rank and full-rank adaptation effectively. In practice it does not seem like LoRA is useful for purposes of model capacity control, only to reduce fine-tuning memory usage.**
> >
> > **A.** In general, our proposed Meta-Adapters objective in Equation 4 does not depend on low-rank adaptation, as it is general to any adaptation method. However, the focus of our analysis is indeed on low-rank adaptation due to its widespread popularity and effectiveness. It has been observed in literature and practice that for model fine-tuning it suffices to simply perturb the model weights by low-rank matrices as it reduces training costs, mitigates catastrophic forgetting, and still achieves strong performance relative to full fine-tuning (i.e. fully retraining the model weights) (Hu 2021, Liu 2024, Dettmers 2023). Inspired by these findings, we showed that incorporating low-rank adaptation in the retraining process gives provable benefits for downstream task adaptation. For these low-rank adaptation methods, the adaptation rank is a hyperparameter and is fixed throughout training. If there existed a LoRA variant with adaptive rank, that could be just as easily incorporated with our Meta-Adapters objective. However, the failure of LoRA to dynamically learn the adaptation rank is orthogonal to our contribution of incorporating fine-tuning methods during retraining using our meta-learning inspired objective.
> >
> > **Q: The model demonstrates no benefit of being useful beyond 3 tasks, despite this presumably being useful in practice. It seems the main purpose of optimization here is parameter recovery rather than statistical learning.**
> >
> > **A.** Thank you for bringing up this distinction. It is correct that our analysis shows no benefit to using more than 3 tasks in the infinite-sample setting in terms of parameter recovery. This was a surprising result, as we mention in the introduction and after the statement of Theorem 3, as this threshold of 3 tasks is independent of the effective task dimension $k$. The focus of our analysis is indeed on parameter recovery, as we do not consider the sample complexity of learning these parameters in the finite-sample setting. In practice with access to finite samples, the obvious benefit to using more tasks is a noise reduction since more samples are incorporated. However, our analysis does not suggest any deeper benefit to using more than 3 tasks for our linear model, which is an interesting and surprising result.
> >
> > Exploring the finite-sample version of this problem, which would better highlight the impact of the number of samples per task and the total number of tasks, is indeed an interesting research direction, and we plan to pursue this in future work.
> >
> > **Q: Code is not provided for the experiments**
> >
> > **A.** Thank you for mentioning this. We have uploaded the code to the supplementary files and will add a link to our repository in the final version.
> >
> > **Q: The abstract describes the FM paradigm as a two-phase procedure but the intro describes it as a three-phase procedure (it seems the abstract does not consider retraining a separate step).**
> >
> > **A.** Thank you for raising this point. In the abstract, we explain that _given_ a pre-trained FM, one typically performs a 2-phase procedure to train the model on a specific task, where phase 1 is retraining and phase 2 is fine-tuning. However, in the introduction we describe the entire pipeline of training an FM from scratch as a 3-stage procedure, where pre-training is the first stage, followed by retraining, and lastly fine-tuning. We have made this distinction clearer by updating the abstract in the current version of the paper.
> >
> > **Q: Remark 1 alludes to the “generation process of each $U\_t^\ast$” but I could not find any description of how those matrices are generated. The same remark also states that “$k(T+1)\ll d$” but I could not find that assumption anywhere (and while perhaps fine for theory, in practice it is not always true).**
> >
> > **A.** Thank you for this comment. We have updated the first paragraph of Section 3 and the remark to explain each $U\_t^\ast \in \mathbb{R}^{d \times k}$ is generated such that each entry is drawn as an i.i.d. Gaussian random variable, where $k \ll d$ and $k(T+1) < d$. Although we only use this for our theoretical results, this assumption may still reasonable hold in practice depending on the model architecture and task structure.

---

> ### Author Response · Authors · 2024-11-28
>
> Thank you again for taking the time to review our work. We just wanted to check if our responses addressed your concerns or if you had any other feedback or suggestions for us.

---

### Official Review · Reviewer_xnyD · 2024-11-03

**Soundness:** 3
**Presentation:** 3
**Contribution:** 2
**Rating:** 6
**Confidence:** 3

**Summary:**

This paper studies the dependency of retraining and fine-tuning in the foundation models. They introduce a meta-learning framework infused
LoRA, namely Meta-LoRA, to fine-tuning for the downstream tasks. They prove that their algorithm can find the second order stationary point and show good performance in ConvAI2 dataset.

**Strengths:**

This paper theoretically shows the standard "Retraining and fine-tuning" method fails to recover the optimal parameters in a low-rank space. Then it proposes a meta-learning framework, namely Meta-LoRA, and prove theoretical guarantee for finding second order stationary point for multi-task fine-tuning.

**Weaknesses:**

Please address the following concerns,

1. This paper proposes a Meta-LoRA method, but uses symmetric low-rank adapters in their method. I am not sure it works in some language models since this restricts the input dimension and output dimension in a layer to be the same, i.e., $U$ and $V$ have the same dimension. For example, not all the layers in RoBERTa satisfy this assumption. The authors also mentioned that they allow asymmetric adapters at test time. If so, I am curious how to get the asymmetric adapters based on the obtained symmetric adapters. So I think this method is somehow limited for most language models in practice. Please address the following questions:

- Clarify how to handle layers with different input and output dimensions in practice.
- Explain in detail how to transit from symmetric adapters during training to asymmetric adapters at test time.
- Discuss any potential limitations or modifications needed for applying the method to common language model architectures.

If they solve these concerns, I'd be willing to increase my rating.

2. The proof of Theorem 3 seems to show $\hat A = A^*$ and $\hat U_t \hat U_t^T  =  U^*_t  (U^*_t)^T $ when $T=3$. How does it hold when $T>3$?

3. This paper construct a contradiction to prove Theorem 4, I think it is the most interesting part in this paper. But it is pretty limited as $T=2$, which means this theory seems unconvincing in practice. So I hope that they can discuss the implications of this limitation for real-world applications where T is typically larger than 2.

4. In experiments, I don't see the report about the number of trainable parameters in your algorithm and SR method. Do both algorithms use the same number of adapters for tasks. If both algorithm tune a specific adapter for a retraining task and use the same number of tuning parameters, I think it is a fair comparison. Please consider the following suggestions:

- Provide a table or detailed description of the number of trainable parameters for each method in the experimental setup.
- Clarify whether the same number of adapters and tuning parameters were used for both algorithms across all tasks.

**Questions:**

Please refer to Weakness.

---

> ### Author Response · Authors · 2024-11-14
>
> **Q: Explain how to deal with asymmetric adapters: clarify how to handle layers with different input and output dimensions in practice, explain in detail how to transit from symmetric adapters during training to asymmetric adapters at test time, and discuss any potential limitations or modifications needed for applying the method to common language model architectures**
>
>  **A:** There are two separate points at play which seemed to have caused your questions. In Section 2, we explain the general Meta-LoRA framework which is ultimately what we use in practice in our LLM experiments in Section 4.2. For this general framework, in equation 5, we state the general Meta-LoRA objective which uses asymmetric adapters $\U_i^{(t)},\V_i^{(t)}$ for the $i$th layer and $t$th task. In terms of the dimensions, we have stated our algorithm in the case where the adapter $\U_i^{(t)}(\V_i^{(t)})^\top$ is a square matrix, as it adapts the model parameter $\W_i \in {\mathbb{R}}^{d \times d}$ which we defined in Section 2.1. We considered adapting square matrices in this section completely for ease of exposition, as our general method will work for adapting any matrix shape, as we simply apply the LoRA method. For example, if we want to apply a rank-$k$ adaptation to $\W \in \mathbb{R}^{d_1 \times d_2}$, we can simply optimize the loss over $\U,\V$ where $\U \in \mathbb{R}^{d_1 \times k}$ and $\V \in \mathbb{R}^{d_2 \times k}$, and $\W$ is adapted as $\W + \U\V^\top$. Thus, our general method has no limitations or modifications needed in order to be applied to general architectures. To be clear about this assumption, in Section 2.1 we added a note that we assume $\W_i$ is square for convenience only.
>
> In your second bullet you referenced our line in Section 3.2 where we mention ''we allow for asymmetric adapters at test time". This statement is only for our theoretical analysis for our simplified linear setting, as this is the only context where we make any restriction to symmetric adapters. Even for our theory, we never construct asymmetric adapters from learned symmetric adapters. We train symmetric adapters on the retraining tasks which leads to the loss defined in Equation 11. From this loss we learn $\Ahat_{\text{Meta}}$, and then when we run LoRA on the test task we allow for the test-task adapters to be asymmetric when perturbing $\Ahat_{\text{Meta}}$. This is shown in our definition of the test loss in Equation 7 which incorporates the asymmetric adapter $\U_{T+1} \V_{T+1}^\top$.
>
> **Q: Does Theorem 3 imply $\hat{A}= A^\ast$ and $\hat{U} \hat{U}^\top = \hat{U^\ast} \hat{U}^{\ast\top}$ for $T>3$**
>
> **A:** Yes, that is correct. Theorem 3 and its proof both imply this result holds for any $T \geq 3$. The proof only relies upon the fact that when $T \geq 3$, there exist three unique indices $r,s,t \in [T]$ such that
>     \begin{equation*}
>         U_r^\ast U_r^{\ast\top} - \hat{U}_r \hat{U}_r^\top = U_s^\ast U_s^{\ast\top} - \hat{U}_s \hat{U}_s^\top = U_t^\ast U_t^{\ast\top} - \hat{U}_t \hat{U}_t^\top.
>     \end{equation*}
> When $T \geq 3$, we can take any set of 3 unique indices and apply the subsequent argument, so for ease of exposition we select $(r,s,t) = (1,2,3)$. This alone then implies the stated result as we show in our proof.
>
> **Q: This paper constructs a contradiction to prove Theorem 4, I think it is the most interesting part in this paper. But it is pretty limited as $T=2$, which means this theory seems unconvincing in practice. So I hope that they can discuss the implications of this limitation for real-world applications where T is typically larger than 2.**
>
> **A:** Theoretically, our result in Theorem 4 only holds for $T=2$. We show numerically that the result is not necessarily true when $T > 2$, and we explain our methodology in Appendix D.2. But, through our linear experiments, we find that even though there may exist problem instances of Equation 11 that are not strict saddle, they do not occur in practice when randomly generating the ground truth parameters. We explain this point in the summary at the end of Section 3 and experimentally verify these claims in our linear experiments in Section 4.1. Further, for our LLM experiments in Section 4.2. we only work in the $T>2$ setting and still show performance benefits of our Meta-LoRA objective. Thus, although the theory for the strict saddle nature of the Meta-LoRA objective is limited to linear models and $T=2$ when considering arbitrary ground truth parameters, this limitation is in theory alone and does not become an issue in practice. We leave it to future work to completely characterize what additional conditions on the ground truth parameters are necessary and sufficient for Theorem 4 to hold for $T>2$. Further, although Theorem 4 does not hold when $T>2$, aspects of our theory show concrete benefits in the $T>2$ setting. For example, Theorem 3 shows that minimizing the loss for linear models when $T>2$ guarantees recovery of the ground truth parameters.

---

> ### Author Response · Authors · 2024-11-14
>
> **Q: In experiments, I don't see the report about the number of trainable parameters in your algorithm and SR method. Do both algorithms use the same number of adapters for tasks? If both algorithm tune a specific adapter for a retraining task and use the same number of tuning parameters, I think it is a fair comparison. Please consider the following suggestions:**
>
> **Provide a table or detailed description of the number of trainable parameters for each method in the experimental setup**
>
>
> **Clarify whether the same number of adapters and tuning parameters were used for both algorithms across all tasks**
>
> **A:** Thanks for bring these points to our attention. For standard retraining, we first learn a single model over the aggregation of the $T$ retraining tasks. For Meta-LoRA, we learn a different fine-tuned models for each of the $T$ retraining tasks, all of which share the same base model weights, so Meta-LoRA uses a small number of additional parameters. After running either of these retraining procedures, the fine-tuning stages are identical and require the same number of trainable parameters no matter which retraining procedure was run. To answer your question for the retraining stage, for simplicity assume our model architecture consisted of $m$ layers, where each layer was parameterized by a $d\times d$ matrix, and we use rank-$k$ adaptations for each layer for our Meta-LoRA objective, where $k \ll d$. Then the SR method uses $md^2$ trainable parameters, while minimizing the Meta-LoRA objective uses $m(d^2 + 2kdT)$ trainable parameters. Since $k$ is small relative to $d$ and we work in the setting where $k(T+1) < d$, asymptotically $m(d^2 + 2kdT) = O(md^2)$ so the increase in trainable parameters is minor. Further, the retraining stage is typically performed offline, whereas we care more about model efficiency in the fine-tuning stage where we want to quickly adapt to new low-resource tasks. We have added these details in Appendix C.1 in the paper.

---

> > ### Comment · Reviewer_xnyD · 2024-11-22
> >
> > I highly appreciate the authors' thorough explanations regarding asymmetric adapters, the limitation of the number of tasks, and computational cost. They have addressed my concerns, so I increased my rating to 6.

---

### Official Review · Reviewer_m5Hr · 2024-11-04

**Soundness:** 3
**Presentation:** 1
**Contribution:** 2
**Rating:** 5
**Confidence:** 3

**Summary:**

This paper presents a novel meta-learning framework for fine-tuning a foundation model to be adaptable to unseen downstream tasks via LoRA fine-tuning. The paper shows for a linear model that standard retraining is suboptimal whereas the proposed method can recover the unique optimal model parameters up to orthogonal symmetry when retraining on three or more tasks. They evaluate the method on a synthetic linear task and a text classification task, finding that Meta-LoRA outperforms standard retraining.

**Strengths:**

- The motivation makes intuitive sense; the standard way we prime models for downstream tasks diverges from how we actually fine-tune them.

**Weaknesses:**

- My main concern is that the experimental evaluation is weak. The paper has one non-synthetic experiment: if I'm understanding correctly, the LLM experiment involves turning the ConvAI2 dataset into a classification task where the model aims to select among possible continuations. There are many standard text classification benchmarks with publicly shared results that would give a better sense of how much Meta-LoRA contributes to downstream performance. Furthermore, there are many substantially better language models than RoBRETa. I'd suggest looking into SmolLM-(135M, 360M) or Qwen-2.5-0.5B in the <1B parameter regime, and there are several other larger models than that which should comfortably fit in a single GPU.
- I found the writing in section 2 to be unnecessarily complicated. For example, section 2.1 describes two stages of fine-tuning with datasets, where the first is over all weights and the second is over LoRA parameters. I think the matrix notation was unnecessary, and the setup shouldn't take over one page to describe.
- You state in the introduction that your framework can be implemented with any PEFT algorithm. Which other PEFT methods do you think the proposed method will work well with?

**Questions:**

Please see weaknesses above.

---

> ### Author Response · Authors · 2024-11-14
>
> 1. **My main concern is that the experimental evaluation is weak. The paper has one non-synthetic exper-
> iment: if I’m understanding correctly, the LLM experiment involves turning the ConvAI2 dataset into
> a classification task where the model aims to select among possible continuations. There are many
> standard text classification benchmarks with publicly shared results that would give a better sense of
> how much Meta-LoRA contributes to downstream performance. Furthermore, there are many sub-
> stantially better language models than RoBRETa. I’d suggest looking into SmolLM-(135M, 360M)
> or Qwen-2.5-0.5B in the <1B parameter regime, and there are several other larger models than that
> which should comfortably fit in a single GPU.**
>
> * Thank you for bringing up your concerns. While our LLM experiments just consider RoBERTa for the ConvAI2 dataset, we would like to highlight that the main contribution and purpose of our paper was to show how incorporating PEFT methods within the retraining process can provably confer performance benefits over standard retraining when adapting to unseen tasks downstream. To our knowledge, this is the first work that shows any strict performance gap between these two methods in any setting. Our experiments mainly serve to verify our theory in practical cases of interest like for the classification task we considered. Ultimately, our experiments served to show the difference between the performance of standard retraining and our Meta-LoRA method, not the absolute performance. Unfortunately, we couldn't find any other relevant datasets for our LLM experiments. If the reviewer could share any other suitable datasets for multi-task learning, we would be more than happy to investigate the relative performance of our method to standard retraining. We are aware of the many publicly available NLP datasets such as GLUE and SuperGLUE, but these are comprised of collections of relatively unrelated tasks, which is not the setting we consider.
>
> 2. **I found the writing in section 2 to be unnecessarily complicated. For example, section 2.1 describes two stages of fine-tuning with datasets, where the first is over all weights and the second is over LoRA parameters. I think the matrix notation was unnecessary, and the setup shouldn't take over one page to describe.**
>
> * Thank you for bringing this to our attention. In this section, we decided to include both the general notation to highlight the flexibility of our framework and then to specifically show how such a framework would work for LoRA, which is the focus of our analysis. In our notation, we consider a model $\Phi$, which we can think about as any standard LLM which maps a sequence of tokens to the predicted next token. In this case, $\mathbf{x}$ would represent the input token sequence, $\mathbf{y}$ would represent the true next token, and $\mathbf{W}$ would be the list of matrices in the attention and feed-forward layers of the model architecture. The fine-tuned model $\Phi_{\text{FT}}$ represents the model for a specific downstream task after fine-tuning, which requires some new parameters $\mathbf{\theta}$. For LoRA, these parameters are the low-rank adapters, but for other fine-tuning methods, they will represent something else. For example, for architecture adapters where new task-specific trainable layers are injected within the network architecture, $\mathbf{\theta}$ represents the parameters of these new layers. We specifically show the representation of $\mathbf{\theta}$ as the low-rank adapters for LoRA in Equation (3) for clarity, as our analysis and experiments ultimately use LoRA as our PEFT method. If the reviewer believes this section must be shortened, we would be happy to do so for the final version.
>
> 3. **You state in the introduction that your framework can be implemented with any PEFT algorithm. Which other PEFT methods do you think the proposed method will work well with?**
>
> * Thank you for highlighting this excellent point. While we focus our analysis on LoRA, our framework can be applied to any PEFT method. Our objective defined in Equation 4 is general to any fine-tuning method, as we simply look to find a value of $\mathbf{W}$ that minimizes the loss on each task after fine-tuning. For example, our method can be infused with architecture adaptations, prefix-tuning, or (soft) prompt-tuning. Specifically for prefix-tuning, $\theta^{(t)}$ would parameterize the prefixes we prepend to the input token sequence at each layer for task $t$. Then using Equation 4, we would jointly minimize the fine-tuned loss on each task over the base model $\mathbf{W}$ which is shared over tasks, and $\theta^{(t)}$, the task specific adapters. An example implementation for the optimization problem in Equation 4 has been added in Appendix E.

---

> ### Author Response · Authors · 2024-11-28
>
> Thank you again for taking the time to review our work. We just wanted to check if our responses addressed your concerns or if you had any other feedback or suggestions for us.

---

### Official Review · Reviewer_Me55 · 2024-11-07

**Soundness:** 1
**Presentation:** 2
**Contribution:** 1
**Rating:** 3
**Confidence:** 3

**Summary:**

This paper proposes a meta-learning approach to learning task-specific models when starting with a pretrained foundation model.  In place of standard retraining (also commonly referred to as continued-pretraining or pre-finetuning) on diverse tasks, a separate set of adapter weights are learned per tasks along with a shared set of global weights.  The shared weights are analogous to weights learned by meta-learning methods like MAML and Reptile that can be quickly adapted downstream to tasks drawn from a task distribution.  The task-specific adapters are taken to be LoRA weights for the theoretical analysis in the linear model setting and the subsequent experiments on a conversational dataset.  In the linear setting, the paper shows that standard retraining and task-specific finetuning is suboptimal relative to the meta-learning approach.  Experiments on the ConvAI2 dataset which models tasks as text from different personas shows meta-learning to outperform standard retraining.

**Strengths:**

- Meta-learning a set of shared global weights for ease of downstream adaptation is a well motivated problem but the writing does not make this motivation clear and the execution is poor especially on the empirical front.
- The performance of meta-learning for better downstream task adaptation is strong on the considered ConvAI2 dataset.

**Weaknesses:**

- The paper note multiple other approaches for meta-learning for foundation models but does not compare to other baselines beyond the retraining + finetuning paradigm.
- There is a lot of work on improving finetuning performance by mixing task specific data with either pretraining data or data from related tasks as the second stage or to replace both stages.  I would expect some of this work to be a baseline in addition to the retraining + finetuning baseline considered.
- It is unclear how the meta-learning stage is conducted.  In particular, are tasks considered one at a time or all mixed together and how are the shared weights $W$ updated?
- The theoretical results in the linear setting are not very useful.  They show that when there are $\geq 3$ tasks, the optimal global parameters can be recovered but that does not tell me what the benefit of increasing additional tasks are.  Typical analysis in meta-learning will consider the task distribution and provide regret bounds that depend on characteristics of the task distribution (for example [1](https://arxiv.org/pdf/1906.02717)).
- Empirical results are limited to just the ConvAI2 dataset when given the limited contributions of other aspects of the paper, I would have expected more of a focus on empirical performance.  Results in Table 1 (b) for rank-8 and rank-16 finetuning are not well explained nor as far as I can tell justified by the theory.

**Questions:**

- Please provide pseudocode for how the meta-learning is conducted.  How does your training approach compare to something like Reptile?
- Why is the performance for rank-16 meta training followed by rank-16 finetuning worse than that for rank-16 meta training followed by rank-8 finetuning?

---

> ### Author Response · Authors · 2024-11-14
>
> **Q: The paper note multiple other approaches for meta-learning for foundation models but does not compare to other baselines beyond the retraining + finetuning paradigm**
>
> **A:** Thank you for raising this point. First, we would like to highlight that the goal of this paper is not proposing the optimal meta-learning framework for training foundation models. The main goal and contribution of the paper is to show that a meta-learning approach can provably help to adapt to unseen downstream tasks relative to standard retraining. To our knowledge, this is the first work that shows a strict performance gap between these two methods in any setting. Our experiments mainly serve to verify our theory in practical cases of interest. We do not compare to other meta-learning strategies as we do not claim that our objective is the optimal meta-learning strategy, but instead show its provable benefits relative to standard non-meta-learning approaches.
>
> **Q: There is a lot of work on improving finetuning performance by mixing task specific data with either pretraining data or data from related tasks as the second stage or to replace both stages. I would expect some of this work to be a baseline in addition to the retraining + finetuning baseline considered.**
>
> **A:** In our paradigm, we assume that we are given a general purpose pre-trained foundation model and want to refine it on a set of tasks of interest so that it can be efficiently adapted to new unseen but related tasks downstream. Given a pre-trained model, we retrain it on a collection of task-specific data. For this stage, we propose our Meta-LoRA objective (Equation 4) rather than the the baseline of standard retraining which just retrains over the aggregation of the task specific data (Equation 1). Then, after either of these retraining strategies, the model is fine-tuned on a single downstream task. This setup seems very similar to the concepts you describe. If there is any difference could you please clarify or elaborate on the methods you are referring to?
>
> **Q: It is unclear how the meta-learning stage is conducted. In particular, are tasks considered one at a time or all mixed together and how are the shared weights $\mathbf{W}$ updated?**
>
> **A:** Thank you for the comment. We define our meta-learning objective function in Equation 4. We have access to all the tasks at once, and look to perform the minimization in (4) over all parameters $\mathbf{W}$ and $\mathbf{\theta^{(t)}}$ for each task index $t=1,\dots,T$. Thus, we neither consider the tasks one at a time (i.e., we do not completely minimize with one task and then move on to the next) nor do we mix all the tasks together (i.e., mix all the data such that we don't know which task the data came from). We have added pseudocode in Appendix E showing a sample implementation of an algorithm for minimizing (4), but we would like to highlight that our contribution is the loss function, not any specific optimization algorithm.
>
> **Q: The theoretical results in the linear setting are not very useful. They show that when there are $T \geq 3$ tasks, the optimal global parameters can be recovered but that does not tell me what the benefit of increasing additional tasks are. Typical analysis in meta-learning will consider the task distribution and provide regret bounds that depend on characteristics of the task distribution (for example 1).**
>
> **A:** We respectfully disagree with the reviewer about the usefulness of our results. Our results consider the structure of the minima and critical points of an infinite-sample population loss. We show that in the infinite-sample regime when $T \geq 3$, the minima of the loss in Equation 11 are the ground truth parameters. Increasing $T$ beyond 3 may confer sample complexity benefits, but we do not consider the finite-sample setting. Further, we construct a generative model with ground truth parameters, so there is no need to do a regret analysis with respect to a hindsight benchmark, as we can directly compare the quality of the learned parameters to the ground truth. Similar types of analysis for other meta-learning algorithms have analyzed the learned parameters relative to a ground truth representation (e.g. [Collins 2022](https://proceedings.mlr.press/v162/collins22a/collins22a.pdf)).
>
> Lastly, our result showing that access to 3 tasks is sufficient for guaranteeing the minimizers of the infinite-sample loss are exactly the ground truth parameters is interesting and surprising. We would like to highlight that this is a positive and counter-intuitive result, especially in contrast to other representation learning results which require the number of tasks to scale according to the effective task dimension ([Collins 2022](https://proceedings.mlr.press/v162/collins22a/collins22a.pdf), [Du 2021](https://openreview.net/pdf?id=pW2Q2xLwIMD) ).

---

> ### Author Response · Authors · 2024-11-14
>
> **Q: Empirical results are limited to just the ConvAI2 dataset when given the limited contributions of other aspects of the paper, I would have expected more of a focus on empirical performance.**
>
> **A:** We respectfully disagree with the reviewer that the theoretical contributions of the paper are limited. To the best of our knowledge, there is no result showing any provable gain for using meta-learning to incorporate PEFT methods for foundation model training. In particular, we prove 4 main theorems. Theorem 1 states that for standard retraining, the recovered solution $\AhatSR$ deviates from the ground truth parameter $\As$ by rank $kT$, so standard retraining never results an an adaptable solution. Theorem 2 shows that the minima of our proposed objective are always low-rank adaptable, as any recovered solution $\Ahat_{\text{Meta}}$ is at most rank $2k$ away from \As. Theorem 3 sharpens this result when $T \geq 3$, as in this case we the only global minima are the ground truth parameters, so $\Ahat_{\text{Meta}} = \As$. Lastly, Theorem 4 states that our proposed loss in Equation 11 is a strict saddle function when $T=2$, so finding a second-order stationary point is sufficient for global minimization. This implies that simple algorithms like perturbed gradient descent can provable achieve global minimization.  Finally, we would like to highlight that since the main focus of our paper is our theory, the experiments serve as a proof of concept to show the benefits of meta learning relative to the standard approach.
>
> **Q: Please provide pseudocode for how the meta-learning is conducted. How does your training approach compare to something like Reptile?**
>
> **A:** Thanks for raising this point. Following your suggestion, we have added the pseudocode of a simple example implementation of the optimization problem for Equation 4 in Appendix E. However, we would like to reiterate that the flexibility of our frameworks lies in the ability to use any optimization algorithm with this new loss.
>
> Thank you for the excellent point about comparing with methods like Reptile. Reptile (and other methods like MAML) implicitly assume that the task-specific fine-tuning performed downstream is a gradient-based method and does not require additional parameters. However, fine-tuning methods like LoRA, prefix tuning, or adapters, often involve new parameters. Reptile as a method is incompatible with these approaches, unlike our method, which is specifically tailored to handle PEFT methods.
>
> **Q: Why is the performance for rank-16 meta training followed by rank-16 finetuning worse than that for rank-16 meta training followed by rank-8 fine-tuning?**
>
> **A:** Thank you for bringing up this point. In practice, we do not have access to the true rank of the adapters needed for meta-learning. Our experiments suggest that a rank of 16 during the retraining phase causes some level of overfitting, as in this phase, we are updating the full model weights $\mathbf{W}$ along with the task-specific adapters. If the adapter rank is too high for this phase, we may not recover an optimally adaptable value of $\mathbf{W}$, especially because the tasks we consider have small numbers of samples relative to the number of parameters of the model.

---

> > ### Comment · Reviewer_Me55 · 2024-11-26
> > **Post author response**
> >
> > I appreciate the additional pseudo code and think it improves the clarity of the paper.  I've increased the score for Presentation to 2.
> >
> > However, I don't feel like my main point around limited experiments, especially given the unrealistic linear setting considered for the theory, was addressed sufficiently.  Hence, I will maintain my score.

---

> > > ### Author Response · Authors · 2024-11-26
> > >
> > > Thank you for reading our response. We would like to address your comment regarding the theoretical result of the paper focusing on an "unrealistic" linear setting.
> > >
> > > This paper aims to pave the way for further theoretical studies on fine-tuning foundation models using ideas from meta-learning; it is not intended to be the final step. There are numerous examples where researchers have developed theory in relatively simple settings such as linear representation learning [1-7] to gather insights for analyzing more complex scenarios. While it is unfortunate that you don't share the same perspective, we respect your point of view.
> > >
> > > [1] Collins et al. (2021). Exploiting Shared Representations for Personalized Federated Learning. *Proceedings of the 38th International Conference on Machine Learning*, in *Proceedings of Machine Learning Research* 139:2089-2099 Available from https://proceedings.mlr.press/v139/collins21a.html.
> > >
> > > [2] Collins et al. “FedAvg with Fine Tuning: Local Updates Lead to Representation Learning”. In:
> > > Advances in Neural Information Processing Systems. Ed. by S. Koyejo et al. Vol. 35. Curran Associates,
> > > Inc., 2022, pp. 10572–10586. url: https://proceedings.neurips.cc/paper_files/paper/2022/file/449590dfd5789cc7043f85f8bb7afa47-Paper-Conference.pdf
> > >
> > > [3] Collins et al. (2022). MAML and ANIL Provably Learn Representations. *Proceedings of the 39th International Conference on Machine Learning*, in *Proceedings of Machine Learning Research* 162:4238-4310 Available from https://proceedings.mlr.press/v162/collins22a.html.
> > >
> > > [4] Tripuraneni et al. (2021). Provable Meta-Learning of Linear Representations. *Proceedings of the 38th International Conference on Machine Learning*, in *Proceedings of Machine Learning Research* 139:10434-10443 Available from https://proceedings.mlr.press/v139/tripuraneni21a.html.
> > >
> > > [5]  Du et al. “Few-Shot Learning via Learning the Representation, Provably”. In: International
> > > Conference on Learning Representations. 2021. url: https://openreview.net/forum?id=pW2Q2xLwIMD
> > >
> > > [6] Thekumparampil et al. “Statistically and Computationally Efficient Linear Meta-representation
> > > Learning”. In: Advances in Neural Information Processing Systems. Ed. by M. Ranzato et al. Vol. 34. Curran
> > > Associates, Inc., 2021, pp. 18487–18500. url: https://proceedings.neurips.cc/paper_files/paper/2021/file/99e7e6ce097324aceb45f98299ceb621-Paper.pdf.
> > >
> > > [7] Chen et al (2022). Active Multi-Task Representation Learning. *Proceedings of the 39th International Conference on Machine Learning*, in *Proceedings of Machine Learning Research* 162:3271-3298 Available from https://proceedings.mlr.press/v162/chen22j.html.

---

### Meta-Review · Area_Chair_9UWA · 2024-12-19

**Metareview:**

The submission proposes a meta-learning approach to adapting foundation models, with the hypothesis that a so-called continued pre-training step that is aware of how the model will be fine-tuned for a particular task will enable better downstream performance. This idea is instantiated by developing a novel continued pre-training and fine-tuning strategy that employs LoRA to control complexity. Theoretical analysis is provided for a simplified linear setting, showing that some benefit is possible. Experimental results are provided on the ConvAI2 dataset and synthetic data.

The reviewers all agree that the idea of meta-learning how to adapt foundation models is well-founded, and the high level approach proposed in this submission has merit. However, all reviewers agreed that the paper could do with stronger theoretical analysis and experimental results.

**Additional Comments On Reviewer Discussion:**

The reviewers did not participate much in the discussion process. However, some of the concerns of reviewer xnyD were addressed, resulting in a raised score.

---

### Decision · Program_Chairs · 2025-01-22

Reject